# Evaluation of Multiple Climate Data Sources for Managing Environmental Resources in East Africa

Solomon H. Gebrechorkos[1,2], Stephan Hülsmann[1], Christian Bernhofer[2]

[1]United Nations University Institute for Integrated Management of Material Fluxes and of Resources (UNU-FLORES), 01067 Dresden, Germany

[2]Faculty of Environmental Sciences, Institute of Hydrology and Meteorology, Technische Universität Dresden, 01062 Dresden, Germany

*Correspondence to*: Solomon H. Gebrechorkos (gebrechorkos@unu.edu)

**Abstract.** Managing environmental resources under conditions of climate change and extreme climate events remains among the most challenging research tasks in the field of sustainable development. A particular challenge in many regions such as East Africa is often the lack of sufficiently long-term and spatially representative observed climate data. To overcome this data challenge we used a combination of accessible data sources based on station data, earth observation by remote sensing, and regional climate models. The accuracy of the Africa Rainfall Climatology version 2 (ARC2), Climate Hazards Group InfraRed Precipitation (CHIRP), CHIRP with Station data (CHIRPS), Observational-Reanalysis hybrid (ORH), and Regional Climate Models (RCMs) are evaluated against station data obtained from the respective national weather services and international databases. We did so by relating point to pixel, point to area grid cell average, and stations average to area grid cell average over 21 regions of East Africa: 17 in Ethiopia, two in Kenya and two in Tanzania. We found that the latter method provides better correlation and significantly reduces biases and errors. The correlations were analyzed at daily, dekadal (10 days), and monthly resolution for rainfall and maximum and minimum temperature (T-max and T-min) covering the period of 1983–2005. At daily time scale, CHIRPS, followed by ARC2 and CHIRP are the best performing rainfall products compared to ORH, RCM, and RCMS. CHIRPS captures well the daily rainfall characteristics such as average daily rainfall, amount of wet periods, and total rainfall. Compared to CHIRPS, ARC2 showed higher underestimation of the total (-30 %) and daily (-14 %) rainfall. CHIRP on the other hand, showed higher underestimation of the average daily

rainfall (-53 %) and duration of dry periods (-29 %). Overall, the evaluation revealed that in terms of multiple statistical measures used on daily, dekadal, and monthly time scale, CHIRPS, CHIRP, and ARC2 are the best performing rainfall products while ORH, individual RCM, and RCMs are the least performing products.

5  For T-max and T-min, ORH was identified as the most suitable product compared to RCM and RCMs. Our results indicate that CHIRPS (rainfall) and ORH (T-max and T-min), with higher spatial resolution, should be the preferential data sources to be used for climate change and hydrological studies in areas of East Africa where station data are not accessible.

# 1. Introduction

In Sub Saharan Africa (SSA) about 80 % of people living in poverty will continue to depend on the agriculture sector as their major income sources under the continuing global change (Dixon et al., 2001; IFPRI, 2009). Unlike in other regions of the world, agricultural activities in SSA are marked by low production mainly due to poor natural resource management, rainfall amount and variability, economy, and technologies. According to IFPRI (2009), reducing poverty in SSA is becoming more challenging due to rapid population growth and associated decline in the quality and availability of environmental resources (e.g. water and soil). Additionally, food security and livelihoods of people are threatened by the direct impacts of change in climate such as increasing frequency of extreme events and weather variability impacts on the production and productivity of agricultural lands (Malo et al., 2012). In general, the impact of climate change in Africa ranges from social and economic to health, water, and food security, which is a threat to the lives of Africans (Urama and Ozor, 2010; Gan et al., 2016).

These outlined challenges hold in particular for the eastern parts of SSA, including Ethiopia, Kenya, and Tanzania. The population (>80 %) mainly depend on agriculture for their livelihood in this region and agriculture-based income contributes 40 % to the country's Gross Domestic Product (GDP) (FAO, 2014). Extreme climate events such as recurring droughts and floods have a tremendous impact on the socio-economy of the region. Devastating droughts in SSA linked to the high variability (seasonal and inter-annual) of rainfall (Sheffield et al., 2013) are projected to increase in frequency (IPCC, 2007, 2014; Niang et al., 2014). In addition to the projected impact, the region is already facing significant food security issues and natural resource-based clashes (UNEP, 2011; World Bank, 2012).

The impacts of future climate change in East Africa vary from region to region. In order to understand the impacts of future climate at the regional and local scale, ground station data with high spatial and temporal resolution is crucial. Regions with poor ground observation are highly vulnerable to climate threats (Wilby and Yu, 2013), which holds particularly for developing countries. In Africa, high quality climate data from meteorological field stations are scarce (Dinku et al., 2013) and inconsistencies exist between other data products largely due to a limited number of ground stations, merging and

interpolation methods (Huffman et al., 2009; Nikulin et al., 2012; Sylla et al., 2013), limited time resolution, and limited documentation quality. In addition, climate data with high temporal and spatial resolution, even if collected by the national meteorological agencies, are often not available due to data sharing policies. With advancements of technologies and research activities, a number of climate data products from different sources (remote sensing, climate model, and reanalysis) have been produced over the last decades that can fill the data gap particularly for drought-prone regions (Gan et al., 2016) and can be used for hydrological and climate change studies.

Several satellite-based rainfall estimates have been developed over the last decades (Sapiano and Arkin, 2009; Zambrano-Bigiarini et al., 2016). In Africa, a list of rainfall and temperature products are available that can be used for climate change studies such as the African Rainfall Climatology Version 2.0 (ARC2) from the Climate Prediction Center (CPC) of the National Oceanic and Atmospheric Administration (NOAA) with a spatial resolution of 0.1° (Novella et al., 2013) and Climate Hazards Group InfraRed Precipitation (CHIRP) and CHIRP with Station data (CHIRPS) from the Climate Hazard Group (CHG) with a spatial resolution of 0.05° (Funk et al., 2015). In addition, the Multi-Source Weighted-Ensemble Precipitation (MSWEP) (Beck et al., 2016), Tropical Applications of Meteorology using Satellite and ground-based observations (TAMSAT) (Tarnavsky et al., 2014), TAMSAT African Rainfall Climatology And Time series (TARCAT) (Maidment et al., 2014), and data from the Enhancing National Climate Services (ENACTS) initiative (Dinku et al., 2014) are available at varying spatial and temporal resolutions and for longer periods.

As another source of climate information, climate model-derived data are suitable tools for assessing climate variability and change. Globally, reanalysis based climate products such as the Observational Reanalysis Hybrid (Sheffield et al., 2006), Modern-Era Retrospective analysis for Research and Applications version 2 (MERRA-2) (Gelaro et al., 2017), and Climate Forecast System Reanalysis (CFSR) (Saha et al., 2010) are widely used for climate and hydrological studies. Moreover, dynamically downscaled data from Global Climate Models (GCMs) is widely used in regional and local scale climate studies. Regional Climate Models (RCMs) produced from dynamically downscaled GCMs provide spatial resolutions that suit end-users (Sun et al., 2006). However, downscaling of climate

information from GCMs to assess the impact of climate change on environmental resources at regional or smaller scale has only recently been performed, e.g. as dynamical downscaling within the CORDEX community (CORDEX-Africa, see e.g. Abiodun et al. (2016)). In Africa (CORDEX-Africa domain) the spatial resolution of RCMs is available at about 0.44º (~50 km) and at varying temporal resolutions. In East Africa, a number of studies have been done with the applications of RCMs for climate studies (Anyah and Semazzi, 2006, 2007; Diro et al., 2011; Endris et al., 2013; Segele et al., 2009).

Before being used as input to different climate or hydrological models, climate data products need to be evaluated against field-based meteorological stations. For studying climate change and climate extremes data with high accuracy and from long periods (> 30 years) are required. In addition, current hydrological (e.g. Soil-Water Assessment Tool, SWAT (Neitsch et al., 2002)) and climate models (e.g. Statistical Downscaling Model (e.g., SDSM, Wilby and Dawson, 2004)) require daily time series of rainfall and temperature covering long periods. Considering these requirements, concerning lengths of time series and temporal resolution on the one hand and the limited availability of station data on the other hand, it is not surprising that comprehensive evaluations of climate data products, particularly on daily time scale, are not available for East Africa to the best of our knowledge. However, few studies are available based on monthly gridded data (e.g., Cattani et al., 2016; Kimani et al., 2017), for limited time periods. Moreover, Kimani et al. (2017) only considered CHIRPS, while a more comprehensive evaluation and comparison of different data sources would be highly desirable. Based on the data requirements of impact models, the climate data products to be included in such an evaluation should be selected/excluded based on high spatial and temporal (e.i., daily) resolution, quality (missing values), and temporal coverage (length of time series), while also taking the results from previous studies (e.g., Cattani et al., 2016; Bayissa et al., 2017; Kimani et al., 2017) in to account.

Therefore, this study aims at comparing and evaluating the available climate data products for Ethiopia, Kenya, and Tanzania at the highest possible spatial and temporal (i.e., daily, for reasons of comparability extended also to decadal and monthly) resolution against station data using the most widely applied and accepted statistical and graphical evaluation methods. Results of our study will help overcome the data scarcity in the study area, in terms of spatial coverage and temporal resolution gaps

of daily, dekadal, and monthly climate data products that can be used for hydrological and climate change and impact studies at a watershed or regional scale. In addition, the data sets can be used for local and regional climate projections using climate models such as Statistical DownScaling Model (SDSM) (Wilby and Dawson, 2004).

## 2. Study area and Data

### 2.1 Study region

The study focuses on the evaluation of daily, dekadal, and monthly climate data sources for regions of East Africa, particularly Ethiopia, Kenya, and Tanzania (Fig. 1). The region is divided by the Great Rift Valley and is topographically one of the most diverse and complex parts of Africa, characterized by multiple rainfall regimes. Generally, the rainfall cycle (climatological annual cycle) in East Africa is linked to the position changes of the Inter-Tropical Convergence Zone (ITCZ) (Endris et al., 2013). Variability in the rainfall patterns in this region is partly induced by local factors such as heterogeneity of land surface and complex topography and their interaction with global climate forcing systems. Countries of the region face similar weather and climate variabilities (spatial and temporal variabilities) and increasing temperature and decreasing precipitation trends (Pricope et al., 2013). In addition, all East African countries face similar issues such as frequent droughts, floods, poverty, and lack of clean and adequate water supply. The conditions can worsen in the near future due to climate change; therefore, sustainable adaptation and mitigation strategies are required, which rely on advanced climate and hydrological models and the respective data inputs.

### 2.2 Data sets

The reference data sets used for evaluation of multiple data products in this study are based on daily rainfall, maximum temperature (T-max), and minimum temperature (T-min) derived from 332 rain gauges and synoptic stations. Station data for Ethiopia was provided by the National Meteorological Agency (NMA) of Ethiopia for the period 1954–2016. The daily data provided by NMA were carefully and extensively checked for its quality and some missing data were infilled from hardcopies. For Kenya

and Tanzania, the global summary of the day available at the National Climate Data Center (NCDC) (https://www.ncdc.noaa.gov/) is used. For evaluation, based on the criteria outlined above, we considered satellite-based rainfall estimates, Observational-Reanalysis Hybrid (ORH), and historical period of Regional Climate Models (RCMs) driven by Global Climate Models (GCMs) to be compared against field-based meteorological stations. The three satellite-based rainfall estimates fulfilling all criteria are the African Rainfall Climatology Version 2.0 (ARC2) (Novella et al., 2013), the Climate Hazards Group InfraRed Precipitation (CHIRP) and CHIRP with Station data version 2 (CHIRPS) (Funk et al., 2015). Not included was, for example, TAMSAT, which is available at higher spatial and temporal resolution and for a longer time period but contains considerable data gaps (Maidment et al., 2017, https://icdc.cen.uni-hamburg.de/1/daten/atmosphere/tamsat-rainfall-africa/) during the evaluation period. ENACTS and TARCAT are available only from dekadal (10 days) time scales. In addition, MERRA-2 and CFS-R are not included in this study due to their coarse spatial resolution compared to the other reanalysis products (i.e., ORH).

ARC2 is the second version of the ARC and is compatible with the algorithm of the Rainfall Estimation Version 2 (RFE 2.0) (Novella et al., 2013). The product is a composite of three hourly geostationary Infrared (IR) data, which makes it different from RFE, centered over Africa provided by the European Organization for the Exploitation of Meteorological Satellites (EUMETSAT) and quality controlled daily rainfall records acquired from the Global Telecommunication System (GTS) gauges. ARC2 is consistent with the historical data sets of the Climate Prediction Center Merged Analysis of Precipitation (CMAP) (Xie and Arkin, 1997) and Global Precipitation Climatology Project (GPCP) (Novella et al., 2013). The data set is updated regularly and it is available at a spatial resolution of 0.1° covering the period from 1983–present.

CHIRPS is a semi-global rainfall product designed for drought monitoring and global environmental changes (Funk et al., 2015). The product provides daily, pentadal, dekadal, and monthly data at a 0.05° spatial resolution available at Climate Hazards Group (CHG ftp://ftp.chg.ucsb.edu/pub/org/chg/products). CHIRPS combines a 0.05° resolution of satellite images and data from ground stations to form a gridded rainfall time series. Station data (see also below) are

used to produce a preliminary two day rainfall product by blending data from sparsely located GTS gauges with rainfall estimates retrieved from the Cold Cloud Duration (CCD) at every pentad. In addition, the final product is developed by blending the best available monthly and pentadal station data with the monthly and pentadal rainfall estimates retrieved from the CCD, respectively, to produce gridded rainfall product (Funk et al., 2015). The development process of CHIRPS includes the 0.05° monthly precipitation climatology (CHPclim), satellite only Climate Hazards Group InfraRed Precipitation (CHIRP), and station blending techniques. The second version of CHIRPS, which is updated regularly, provides an improved daily rainfall time series (1981–present) with a spatial resolution of 0.05° ranging from 50ºS to 50ºN (and all longitudes) (Funk et al., 2015). The development process of CHIRPS and its application in drought monitoring in Africa (e.g. Ethiopia) is explained in detail by Funk et al. (2015). CHIRPS is not only used for drought monitoring, but also for other global environmental applications (Zambrano-Bigiarini et al., 2016), water resource management, and climate dynamics (Ceccherini et al., 2015; Deblauwe et al., 2016). ORH is a global (Sheffield et al., 2006) and regional (Northern/West/East Africa) (Chaney et al., 2014) three-hourly, daily, and monthly meteorological data set covering the period between 1901–2012. ORH is developed by a spatial downscaling of the NCEP–NCAR reanalysis (Kalnay et al., 1996) up to a spatial resolution of 0.1º using a bilinear interpolation. ORH merges multiple data products such as the NASA Langley Surface Radiation Budget (SRB),the monthly temperature data from the University of East Anglia Climate Research Unit (CRU), Tropical Rainfall Measuring Mission Multi-satellite Precipitation Analysis (TMPA) (Huffman et al., 2007) and other observational based rainfall products (Chaney et al., 2014). The spatial downscaling of ORH is done with the inclusion of changes in elevation and it is evaluated against ground stations (global summary of the day) available at the US National Climatic Data Center (NCDC). ORH is corrected for temporal inhomogeneity and biases and random errors are omitted through assimilation with quality controlled and gap filled ground station data available at NCDC (https://www.ncdc.noaa.gov/) as a global summary of the day (Chaney et al., 2014). This data is freely available from the Terrestrial Hydrology Research Group, University of Princeton (http://hydrology.princeton.edu). Even though ORH is not updated regularly, it has been widely used in

climate and hydrological studies (e.g., Troy et al., 2011; Wang et al., 2011; Demaria et al., 2012; Sheffield et al., 2014)

Compared to the other rainfall products, monthly ground station data from Ethiopia, Kenya, and Tanzania are included in CHIRPS. Evaluating CHIRPS based on ground station data might thus raise concerns about the independence of data. However, not all stations used in this study are included in CHIRPS and in general the stations are not consistently used in the development process of CHIRPS. In addition, the station data used in CHIRPS is mainly a monthly total from a limited number of stations. For example, in Ethiopia, where all station data originate from NMA, during 01/1983–02/1983 and 08/2005–12/2005 the monthly stations used in CHIRPS declined from 139 to 132 and from 175 to 169, respectively. In 2015, the number of station included in CHIRPS even declined to below 10. Moreover, in Kenya and Tanzania, during the period of 01/1983–12/2005 the number of station used in CHIRPS declined from 142 to 62 and from 171 to 55, respectively (ftp://chg-ftpout.geog.ucsb.edu/pub/org/chg/products/CHIRPS-2.0/diagnostics/). Besides the difference in temporal resolution (monthly vs. daily) and the number of stations between station data included in CHIRPS and the validation data set, the latter deviated from the former since we used original data provided by NMA (Ethiopia) which were quality-controlled and extended by adding data from hard copies. Overall, while not fully independent, the relation between CHIRPS and the validation data set should be weak, besides the fact that there is no other (fully independent) validation data set available.

Historical data (control model runs) of the CORDEX RCMs are also used as a potential source for rainfall, T-max, and T-min data. RCMs are climate models with a higher spatial resolution compared to GCMs. The driving data of RCMs are derived from GCMs or reanalysis data and can include greenhouse gases (GHG) and aerosol forcing. Compared to GCMs, RCMs considers local factors such as complex topography and land cover inhomogeneity in a physically based manner (IPCC, 2007). In Africa, dynamical downscaling was performed in a large effort within the CORDEX community (CORDEX-Africa). Within CORDEX-Africa the continent's climate was dynamically modelled by an international consortium, providing a spatial resolution of about 50 km. According to the IPCC report (2007), RCMs can be used for wide range applications such as climate change studies. Following the

recommendation of Endris et al. (2015), the historical data derived from two CORDEX RCMs, RCA (Samuelsson et al., 2011), and COSMO-CLM or CCLM (Baldauf et al., 2011), driven by HadGEM2-ES (MOHC, United Kingdom), MPI-ESM-LR (MPI, Germany), and GFDL-ESM2M (NOAA/GFDL, United States) are used. Rainfall, T-max, and T-min products of both RCMs are retrieved from the Earth System Grid Federation (ESGF) data portal.

## 3. Methodology

### 3.1 Selection of validation areas and ground stations

The evaluation of multiple daily, dekadal (10 days), and monthly rainfall, T-max, and T-min products were conducted on selected basins of Ethiopia (EthioShed1 - EthioShed17), Kenya (KenShed1 and KenShed2), and Tanzania (TanzShed1 and TanzShed2) (Fig. 1). The polygons in Fig. 1 are river basins retrieved from the global river basins available at the WaterBase hosted by the United Nations University (UNU-INWEH: http://www.waterbase.org/). In most regions of Africa not only are the density and availability of field-based meteorological stations limited, but their accessibility is very restricted for many reasons. For this study, it was only possible to get daily station data from the National Meteorological Agency (NMA) of Ethiopia with a reasonable spatial and temporal coverage. Therefore, the selection of validation areas is based on the availability, quality, and density of field-based meteorological stations during the period of 1983–2005. It was almost impossible to find multiple stations in one satellite grid cell. For Kenya and Tanzania, therefore, stations with more than 10 years (>50 % of the study period), were included for evaluation (Table 1).

The quality of selected stations was checked and extremely high rainfall records during dry seasons, such as daily rainfall of > 480 mm preceding and following by dry days, were excluded. Finally, a total of 132 stations were found suitable for comparison, 2 to 12 stations located in the validation areas. In addition to these stations in the validation areas, 78 stations, randomly distributed over the region, are used to compare on an individual basis with the rainfall and temperature products. Compared to Kenya and Tanzania, the quality, continuity, and spatial and temporal coverage of stations were better in Ethiopia and only stations with missing values of less than 20 % were considered. The availability of

multiple stations in a validation area helps to check the quality of individual stations by using methods such as double mass curve (Vernimmen et al., 2012) and allows for replacement of missing values of one station from a nearby station.

## 3.2 Comparing ground data with satellite, observational reanalysis, and climate model-based data

The most commonly used method to compare ground observations with other data products such as satellite based rainfall estimates and climate model outputs is point (station) to pixel comparison. When comparing daily rainfall, particularly in very complex topography, on point to pixel basis it can be challenging to acquire reasonable agreements. Therefore, in this study we used point to pixel, point to area grid cell average, and stations average to area grid cell average to evaluate the accuracy of each product. Area grid cell average is the average of numbers of pixels covering the basin or the validation area. Similarly, station average is used to indicate the average value of the stations inside the validation area. Therefore, during the comparison process, individual or the stations average is compared to the area grid cell average of the product. The most commonly used statistical methods such as the Pearson correlation coefficient (CC), bias, relative bias (Rbias), Mean Absolute Error (MAE), Root Mean Square Error (RMSE), and Index of Agreement (IA) (Cohen Liechti et al., 2012; Daren Harmel and Smith, 2007; Moazami et al., 2013) are used. CC (Eq. 1) is applied to evaluate the agreement of individual products (P) to station data (O). A value of CC close to one shows a perfect positive fit between the products and station data.

$$CC = \frac{\sum_{i=1}^{N}(P_i - \bar{P}) \cdot (O_i - \bar{O})}{\sqrt{\sum_{i=1}^{N}(P_i - \bar{P})^2} \cdot \sqrt{\sum_{i=1}^{N}(O_i - \bar{O})^2}} \tag{1}$$

The average differences and systematic bias of each product are given as bias (Eq. 2) and Rbias (Eq. 3). Bias can be positive (overestimation) or negative (underestimation) according to the accuracy of each product.

$$\text{Bias} = \frac{\sum(P_i - O_i)}{N} \tag{2}$$

$$\text{Rbias} = \frac{\sum_{i=1}^{N}(P_i - O_i)}{\sum_{i=1}^{N} O_i} \times 100 \tag{3}$$

The MAE and RMSE (Eq. 4 and 5), are well-known and accepted indicators of goodness of fit that shows the differences between ground observation and model or other product outputs (Legates and McCabe, 1999).

$$MAE = \frac{\sum_{i=1}^{N}|O_i - P_i|}{N} \tag{4}$$

$$RMSE = \sqrt{\frac{\sum_{i=1}^{N}(O_i - P_i)^2}{N}} \tag{5}$$

The IA (Willmott, 1981) is another widely used indicator of goodness of fit between observed and model output. IA (Eq. 6) describes how much of the model or product output (rainfall, T-max, and T-min products) are error-free compared to the ground observations.

$$IA = \frac{\sum(P_i - O_i)^2}{\sum(|P - \bar{O}| + |O - \bar{O}|)} \tag{6}$$

In addition to the above statistical methods, the Taylor diagram (Taylor, 2001) is used to summarize the statistical relationship between ground station data and the products for rainfall, T-max, and T-min. In this diagram, the relationships between the two fields are explained by the correlation coefficient (R), centered mean square (RMS) difference (E´), and standard deviation (σ). The diagram is useful for evaluating the accuracy of multiple data sources or model output against a reference or observational data (IPCC, 2001). A single point on the diagram displays three statistical values (R, E´, and σ) and their relationship is given by Eq. (7).

$$E'^2 = \sigma_f{}^2 + \sigma_r{}^2 - 2\sigma_f\sigma_r R \tag{7}$$

Where $\sigma_f^2$ and $\sigma_r^2$ are the variance of the model and observation fields and R is the correlation coefficient between the two fields (Eq. 8).

$$R = \frac{\frac{1}{N}\sum_{n=1}^{N}(f_n-\bar{f})(r_n-\bar{r})}{\sigma_f\sigma_r} \tag{8}$$

In the diagram, the distance from the reference point (observed data) is given as the centered RMS
difference of the two fields (Eq. 9). A model with no error would show a perfect correlation to the observation.

$$E'^2 = \frac{1}{N}\sum_{n=1}^{N}[(f_n-\bar{f})-(r_n-\bar{r})]^2 \tag{9}$$

Where $f$ is the test (e.g. model or satellite) field and $r$ is reference (observed) field, whereas $\sigma_f\sigma_r$ are the standard deviations of the model and reference fields (Eqs. 10 a and b).

$$\sigma_f = \sqrt{\frac{1}{N}\sum_{n=1}^{N}(f_n-\bar{f})^2} \tag{10a}$$

$$\sigma_r = \sqrt{\frac{1}{N}\sum_{n=1}^{N}(r_n-\bar{r})^2} \tag{10b}$$

Additionally, rainfall characteristics such as the number wet days, duration and amount of wet periods, duration of dry periods, and daily and total rainfall are used to evaluate the accuracy of individual rainfall products by comparing to the observed data. Rainfall characteristics are widely used indicators
in rainfall modelling (Wilby and Dawson, 2007; Jebari et al., 2012) and include: Number of wet days (days $y^{-1}$), which is the count of days with rainfall per year; duration (days) of wet and dry periods indicating the average number of consecutive wet and dry days during the study period; the amount of wet periods (mm), indicating the amount of rainfall observed during the identified wet period.

# 4. Results

## 4.1 Validation of satellite, observational reanalysis, and climate model-based products

The average daily rainfall (Fig. 2) of the study region retrieved from ARC2, CHIRP, CHIRPS, ORH, individual RCMs (RCM) and RCMs mean (RCMs) displays large discrepancies between the products for the study period 1983–2005. Compared to dekadal and monthly resolution, the comparison at daily time scale, particularly of rainfall, is challenging and more emphasis is given on this evaluation. RCMs (RCA4 and CCLM) driven by HadGEM2-ES (HadGEM2), MPI-ESM-LR (MPI), and GFDL-ESM2M (GFDL) are used in this study. For RCMs driven by each GCM, the average is used. The daily rainfall, T-max, and T-min maps of GFDL display the result of single RCM (RCA4) driven by GFDL-ESM2M for the period of 1983-2005. Higher and lower average daily rainfall values are displayed by GFDL and ORH, respectively (Fig. 2). However, all the products showed a similar tendency in capturing the daily rainfall distribution; higher in the western and lower in the eastern part of the region. In addition, the average daily T-max and T-min (Fig. 3) of the region shows relatively higher disagreement between ORH and individual RCMs (RCM). However, RCM shows a higher agreement in Ethiopia, Kenya, and Tanzania for T-max and T-min.

The relation of each product with station data is given by scatter plots in Fig. 4 for eight validation areas, four in Ethiopia, two in Kenya, and two in Tanzania. The same plots – with similar results - for another 13 areas in Ethiopia are provided in the supplementary material (SF. 1). The monthly rainfall plots display the relationship of each product with observed ground data and this relation is explained by the coefficient of determination or R-Squared ($R^2$). Based on the scatter plots, CHIRPS and CHIRP are the most accurate rainfall products, with higher correlation and lower RMSE, and ARC2 and ORH are the second best products. RCM and RCM's mean (RCMs) correlate weakly in most of the validation areas. In addition, RCM (not shown in Fig. 4) and RCMs show a strong over- and underestimation of monthly rainfall compared to the other products. In EthioShed1, for example, CHIRPS and CHIRP are shown to be the most accurate products, while ARC2 and ORH showed higher dispersion above and below the regression line. Similarly, in EthioShed4 both CHIRP and CHIRPS have an equal R2, but in

terms of biases (points below and above the regression line) CHIRPs performed better. The observed biases in CHIRP and higher correlations in CHIRPS in all the validation areas highlights the role of the station-satellite data blending techniques. Compared to other validation areas, the agreement of products in EthioShed16 is comparably weak and CHIRPS and CHIRP showed the higher $R^2$ (0.48) compared to ARC2, ORH, and RCMs.

As for the daily, dekadal, and monthly resolution, the comparison is performed in three ways: point to pixel, point to area grid cell average, and stations average to area grid cell average using the methods described in Section 3.2. An explanatory example is given in Table 2, using stations of EthioShde1 displaying the difference in comparing products through point (station) to pixel, point to area grid cell average, and stations average to area grid cell average. The agreement of each product with station data on a daily time scale and on point to pixel comparison is weak, with significantly higher biases and errors. For rainfall, in general, the latter method, stations average to area grid cells average, provides better correlation, higher index of agreement, and lower biases and errors. Compared to point to pixel, the stations average to area grid cells average improves the correlations of ARC2, CHIRP, and CHIRPS by 81.3 %, 65.7 %, and 8 %, respectively. In addition to the correlation, the method reduces the RMSE by more than 66 %. Compared to ARC2 and CHIRP (Table 2), CHIRPS gives a significantly higher correlation and IA and lower biases and RMSE. During area averaging, extremely high rainfall events obtained for a location from the various data products are levelled off by averaging and this makes the product more representative for the area. In most of the rainfall products, there are occasionally higher daily rainfall values recorded and the averaging removes those extremes, which are much higher than the observed station data in the area. Compared to point to pixel, the second method, point to area grid cell average, provides a reasonable correlation.

The agreement of each product increases with decreasing temporal resolution, from daily to dekadal and monthly resolutions. Including the historical data of each RCM, RCMs, and ORH, the overall comparison using some of the statistical methods is summarized in Tables 3, 5, and 6 for rainfall, T-max, and T-min, respectively. The evaluation of each rainfall product (ARC2, CHIRP, CHIRPS, ORH, and RCMs) showed a different degree of agreement with station data (Table 3). The same table for

individual RCMs (RCM) for all the validation areas is provided in the supplementary material (ST 1). At daily time scale, CHIRPS followed by ARC2 and CHIRP proved to be the most accurate rainfall products compared to ORH, RCM, and RCMs in all the validation areas. In general, out of the 21 validation areas CHIRPS, ARC2, and CHIRP showed a higher correlation in 17, 3, and 1 validation areas, respectively. In addition to the higher correlation, CHIRPS, CHIRP, and ARC showed lower RMSE than ORH, RCM, and RCMS. Similarly, CHIRPS and CHIRP showed lower biases than observed in ARC2, ORH, RCM, and RCMs in most of the validation areas.

On average, over the 21 validation areas, CHIRPS captures well the number of wet days (-0.17 % deviation), average duration of wet (-13.4 % deviation) and dry periods (-17.6 % deviation), total rainfall (-4.5 % deviation), average amount of wet periods (-17 % deviation), and average daily rainfall (-7.7 % deviation) (Table 4). Next to CHIRPS, ARC2 showed higher agreement in producing the average duration of wet (-20 % deviation) and dry periods (11.3 % deviation) and average amount of wet periods (-38 % deviation). CHIRP, on the other hand, showed a higher agreement in the total amount of rainfall with a 2.75 % deviation, which is higher than CHIRPS, ARC2, ORH, RCM, and RCMS. On the contrary, ARC2 and GFDL showed higher under- (-34.7 % deviation) and overestimation (27.2 % deviation), respectively, of the total amount of rainfall compared to the other products. In addition, ARC2 showed a higher underestimation in number of wet days (-15.1 % deviation) and average daily rainfall (-23.8 % deviation) compared to CHIRPS and ORH. CHIRP, on the other hand, showed higher overestimation in number of wet days and duration and amount of wet periods (> 59.7 % deviation) and underestimates duration of dry periods and average daily rainfall (-62 % deviation) compared to the other products. Moreover, RCMs, next to CHIRP, showed higher overestimation in number of wet days and duration and amount of wet periods (> 44.9 % deviation) and total rainfall amount (11.4 % deviation) and underestimate average duration of dry periods and daily rainfall by about 41 %. In general, the observed rainfall characteristics are well captured by CHIRPS, with a percentage difference from the observation of -0.17 % to -17.6 % for number of wet days and duration of dry periods, respectively, compared to CHIRP, ARC2, ORH, RCM, and RCMs (Table 4).

For T-max and T-min, only ORH, RCM, and RCMs are compared with station data. For 21 validation areas ORH data proved to be the most accurate product for both T-max (Table 5) and T-min (Table 6). In comparison to RCM and RCMs, ORH showed a significantly higher correlation and lower biases and errors in most of the validation areas. In seven of the 21 validation areas, RCMs showed a higher correlation in T-max than ORH and RCM. However, for T-min, ORH in 20 of the 21 validation areas showed a higher correlation. In general, RCM and RCMs showed higher RMSE and biases in most of the validation areas compared to ORH. Next to ORH and compared to RCM, RCMs appeared to be the best data source particularly for T-max. RCMs showed a relatively higher correlation and lower biases and errors compared to RCM in most of the validation areas.

## 4.2 Validation of satellite, observational reanalysis, and climate model-based products at dekadal and monthly resolutions

To understand the role of higher spatial resolution at improving the agreement with station data, a similar statistical evaluation was performed using the coarse resolution of CHIRPS (0.25°). Compared to the coarse resolution of CHIRPS, the daily improved version (0.05°) used in this study showed an increased correlation of up to 3.2 % in all the validation areas. In line with the daily evaluation, the comparison was extended to dekadal and monthly resolutions for rainfall, T-max, and T-min using the same statistical methods. For this analysis the observed daily ground observations and data from ARC2, CHIRP, CHIRPS, ORH, RCM, and RCMs were aggregated to dekadal and monthly resolutions. With decreasing temporal resolution (daily to monthly), the agreement of each product showed a marked improvement in all the validation areas. In addition to the increase in correlation, biases (bias and Rbias) and errors (MAE and RMSE) in rainfall are decreased at dekadal and monthly resolutions.

At dekadal and monthly resolution, the agreement of all rainfall products with station data increased compared to daily resolutions and the result for eight validation areas of Ethiopia, Kenya, and Tanzania are given in Fig. 5. The same plots – with similar results - for another 13 areas are provided in the supplementary material (SF. 2). Similar to the daily evaluation, CHIRPS appeared to be the most accurate rainfall product both at dekadal and monthly resolutions in most of the validation areas

compared to the other products. In addition to the higher correlation of CHIRPS with station data at monthly and dekadal time scale, the centered mean square (RMS) difference and standard deviation is close to the observation in most of the validation areas. Following CHIRPS, CHIRP appeared to be the second best data source for dekadal and monthly rainfall and in three validation areas (EthioShed3, 15, and 16) showed a slightly higher correlation than CHIRPS. In two validation areas (KenShed1 and 2), ARC2 showed a slightly higher correlation than CHRIP and CHIRPS. However, in KenShed2 ARC2 showed a higher deviation from the observed value compared to CHIRP and CHIRPS. CHIRPS has, for example, almost similar standard deviation as the station data in all the validation areas except in areas with lower number of ground stations (EthioShed12–15 and TanzShed1). Overall, CHIRPS, CHIRP, and ARC2 were found to be the best performing rainfall product while ORH, RCM, and RCMs are the least performing products.

Moreover, for T-max and T-min, the correlation of ORH, RCM, and RCMs increased from daily to dekadal and monthly resolutions. The agreement of each product with station data, for eight validation areas of Ethiopia, Kenya, and Tanzania, is given in Fig. 6 and Fig. 7 for T-max and T-min, respectively. The same plots – with similar results - for another 13 areas are provided in the supplementary material (SF. 3 and SF. 4 for T-max and T-min, respectively). Compared to RCM and RCMs, the correlation between ORH and station data is higher in most of the validation areas. In addition, ORH showed lower centered mean square (RMS) difference and biases (bias and Rbais). In addition, compared to the RCM and RCMs the standard deviation of ORH is close to the respective observations in most of the validation areas. Compared to RCM, the standard deviation and centered mean square (RMS) difference of RCMs is lower in most of the validation areas.

## 5. Discussion

Detection of rainfall characteristics by satellite observations or climate model simulations' output (GCM and RCM) is very challenging as compared to temperature. This is especially evident in East Africa, where the topography is complex and characterized by multiple rainfall regimes. In particular, it is difficult to estimate rainfall with satellite imageries in the mountainous region of East Africa (Cattani

et al., 2016) because these products are inevitably not representing the regional rainfall patterns and complexity of the region's topography (Romilly and Gebremichael, 2010). Here, for an improved understanding of the climatic condition of this complex region and its impact on environmental resources, daily rainfall, T-max, and T-min products from high resolution satellite imageries, observational-reanalysis, and climate models outputs are compared against ground observations. Such an evaluation was not available as of yet for the considered region. Therefore, an in-depth evaluation was performed, particularly on a daily time scale, of the satellite-based rainfall products (ARC2 CHIRPS and CHIRP), ORH, and RCMs (CCLM and RCA) driven by three GCMs. ARC2, CHIRP, and CHIRPS are rainfall products, whereas ORH and RCMs provide rainfall, T-max, and T-min.

From the comparison (using point to pixel, point to area grid cell average, and stations average to area grid cell average), the stations average to area grid cell average showed the best correlation and least biases and errors in all the validation areas. A study by Duan et al., (2016) in Adige Basin (Italy) found that comparing rainfall products such as CHIRPS on a watershed scale showed a marked improvement in overall agreement compared to point to pixel on daily and monthly time scale. Comparing the coarse resolution of satellite products and of RCMs using the point to pixel method cannot be expected to result in a high agreement with station data. Ground stations provide point data measured over continuous time periods, whereas satellite products provide area averages based on discontinuous (rain) estimates. Field-based stations (as point measurements) cannot be considered as reference data for evaluation of area-based rainfall estimates (Cohen Liechti et al., 2012; Wang and Wolff, 2010), if not compared at a monthly or annual time scale. This is similar to our finding that the point to pixel comparison for all products inside and outside the validation areas show weak statistical relations with ground stations (e.g. see Table 2). The correspondence of all products at a daily time scale and in all the validation areas was found comparably weak and the findings are in agreement with earlier studies (Cohen Liechti et al., 2012; Dembélé and Zwart, 2016).

At daily time scale, CHIRPS followed by ARC2 and CHIRP showed higher correlation and lower errors and biases in all the validation areas compared to ORH, RCM, and RCMs. In addition, CHIRPS captures the daily rainfall characteristics well while ARC2 showed higher underestimation of the

average daily and total rainfall. The agreement of all the rainfall products increases from daily to dekadal and monthly time scale (Fig. 5) and this is consistent with other studies (Cohen Liechti et al., 2012; Dembélé and Zwart, 2016; Kimani et al., 2017). Generally, CHIRPS with high spatial resolution, followed by CHIRP and ARC2, was the best performing rainfall product in terms of correlation, biases and errors and in characterizing regional rainfall characteristics. By contrast, ORH, RCM, and RCMs appeared to be less precise rainfall products at all time scales and in all validation areas. When looking at the performance of different data products in the selected validation areas (Fig. 4), dispersion is comparably higher in areas with lower number of ground stations. An additional confounding factor could be the very complex topography of the region. This might explain why products with coarser spatial resolution (ORH, RCM, and RCMs) showed higher dispersion compared to products with higher spatial resolution (CHIRPS, CHIRP, and ARC2).

The daily rainfall data (global summary of the day) available at the National Climate Data Center (NCDC) needed to be controlled for quality before application. In East Africa, particularly Ethiopia, the available data at NCDC is very poor and only few stations are available. Therefore, products developed based on the global summary of the day such as ORH cannot be expected to provide accurate results particularly for the most complex climate variable, rainfall, as CHIRPS and ARC2. CHIRPS incorporates monthly station data obtained from different regional meteorological organization, e.g., from Ethiopia, Kenya, and Tanzania. In all the validation areas one to seven stations were included in the development of CHIRPS in different months during 1981–2005. In EthioSded1 (Table 2), for example, six of the nine stations we considered in this study are included in CHIRPS. The inclusion of monthly station data can be assumed to improve CHIRPS´ performance compared to other rainfall products. This particular feature of CHIRPS (compared to CHIRP and other data products) is somewhat problematic for our analysis, since the correlated data are not fully independent. However, since only monthly data from a limited number of stations were included in CHIRPS, the dependency is rather weak and indirect. In fact, the improved performance of CHIRPS was shown even in areas were station data is not included (e.g. Arijo, Bedele, and Hurma stations in EthioShed1) and on daily time scale.

Even though ORH was one of the least performing rainfall product, it appeared to be the most accurate data source for T-max and T-min at daily, dekadal, and monthly resolutions compared to RCM and RCMs. Nikulin et al., (2012) presented a detailed comparison of daily gridded observations with multiple RCMs including RCA and CCLM and they found large discrepancies over the whole region of Africa. However, in this region, RCMs appeared to be the second best data source for both T-max and T-min and RCM are less precise with slightly higher biases and errors. In this region, other studies (Endris et al., 2013; Kim et al., 2014) concluded that the multi-model or ensemble mean of CORDEX RCMs provides reasonable results compared to individual RCMs (RCM). The systematic bias of RCM and RCMs is higher in most of the validation areas compared to the other products, particularly for rainfall, that can be improved by applying different bias correction techniques such as the empirical quantile mapping (Lafon et al., 2013; Maraun, 2013; Teng et al., 2015) before application to different hydrological and climate models (e.g., SDSM). In general, in topographically complex regions such as East Africa, RCMs require further improvements in terms of spatial resolution and accuracy by adding more local information in the modelling process, particularly for precipitation.

## 6. Summary and Conclusion

The evaluation of rainfall, T-max, and T-min from different sources against station data was performed for large parts of East Africa (Ethiopia, Kenya, and Tanzania) using three methods: point to pixel, point to area grid cell average, and stations average to area grid cell averages. Compared to the other two methods the latter method (stations average to area grid cell average) provides a better correlation and index of agreement (IA) and lower errors (MEA and RMSE) and biases (bias and Rbias). Using this method, individual rainfall, T-max, and T-min products were compared at daily, dekadal (10 days), and monthly resolutions. At daily time CHIRPS, ARC2, and CHIRP provide a better agreement with station data compared to ORH, RCM, and RCMs. Compared to CHIRPS and CHIRP, ARC2, ORH, RCM, and RCMs showed higher biases and errors in most of the validation areas. Overall, the performance of CHIRPS is higher than the other rainfall products in capturing the daily rainfall characteristics such as number of wet days, duration of wet and dry periods, total and daily rainfall, and amount of wet periods. ARC2 better captures duration of wet and dry periods, but showed higher underestimation of the total

and daily rainfall and number of wet days compared CHIRPS and CHIRP. RCM and RCMs, on the other hand, showed higher overestimation in number of wet days, duration and amount of wet periods, and total rainfall and underestimate average duration of dry periods and daily rainfall.

ORH, on the contrary, appeared to be one of the least-performing rainfall products for the study region, but the most accurate product, compared to RCM and RCMs, for T-max and T-min at daily time scale in most of the validation areas. The evaluation of the above products at dekadal and monthly time scales showed that CHIRPS with high spatial resolution (0.05°) has higher correlation and lower errors and biases than the other rainfall products. As the temporal resolution gets coarser (e.g. monthly), the correlation between ground observation and the above products significantly increases. In addition, biases (bias and Rbias) and errors (MAE and RMSE) significantly decreased. Similar to that of rainfall, the comparison at dekadal and monthly resolution showed an improved correlation and lower errors and biases for both T-max and T-min. Compared to RCM and RCMs, ORH with higher spatial resolution was found to be more accurate at dekadal and monthly resolutions. Next to ORH, RCMs showed a better performance than RCM, with lower biases and errors.

In general, CHIRPS for rainfall and ORH for T-max and T-min performed best in the considered regions of Ethiopia, Kenya, and Tanzania. Further studies need to confirm whether this finding holds for other regions as well and our approach may represent a blueprint how to address this question. Since CHIRPS and ORH are available with higher spatial and temporal resolution and for longer periods, these data sources can be used for long-term climate studies (trend, variability, and extreme indices) and input for climate or hydrological models. Considering the typical need for daily data for model input, it remains to be investigated whether poor daily data with a limited bias and similar variance are an acceptable replacement of missing station data when used for impact model studies. In addition, the products can be used to check the plausibility of available ground stations or substitute ground observation in regions of Ethiopia, Kenya, and Tanzania where ground station data are not available or accessible.

Table 1: General characteristics of selected validation areas and meteorological stations covering the time period 1983–2005.

| Validation areas/basins | Basin area (km$^2$) | Average area elevation (m) | Number of stations | Average station elevation (m) | Average annual rainfall (mm) | Average T-max /T-min |
|---|---|---|---|---|---|---|
| EthioShed1 | 8980 | 1516 | 9 | 1881 | 1758 | 26.3/13.6 |
| EthioShed2 | 12828 | 2279 | 12 | 2009 | 968.6 | 25.9/11.4 |
| EthioShed3 | 15123 | 2192 | 9 | 2104 | 1202.6 | 25.3/11.7 |
| EthioShed4 | 8323 | 2180 | 7 | 1954 | 994.42 | 31.8/16.5 |
| EthioShed5 | 5625 | 1720 | 10 | 1800 | 1039.1 | 26.4/13.4 |
| EthioShed6 | 11204 | 2830 | 8 | 2510 | 1168.7 | 22.1/8.0 |
| EthioShed7 | 12445 | 1830 | 8 | 1973 | 1524.53 | 25.7/12.4 |
| EthioShed8 | 6522 | 1930 | 5 | 2022 | 1628.35 | 26.0/14.0 |
| EthioShed9 | 4666 | 1526 | 4 | 1738 | 578.4 | 28.0/14.4 |
| EthioShed10 | 5986 | 2520 | 8 | 2580 | 1133.1 | 21.2/9.3 |
| EthioShed11 | 11496 | 1256 | 7 | 1468 | 945 | 27.4/15.2 |
| EthioShed12 | 3868 | 520 | 2 | 400 | 343.8 | 34.1/22.3 |
| EthioShed13 | 4934 | 1301 | 4 | 2413 | 588 | 26.2/13.1 |
| EthioShed14 | 2835 | 1360 | 4 | 1239 | 706 | 31.8/16.5 |
| EthioShed15 | 1121 | 2307 | 4 | 2183 | 495 | 24.3/11.1 |
| EthioShed16 | 3012 | 2102 | 5 | 2148 | 1110 | 26.0/11.8 |
| EthioShed17 | 9909 | 1998 | 12 | 2056 | 2075 | 23.8/10.2 |
| KenShed1 | 11712 | 1980 | 4 | 1024 | 1156.1 | 25/13.5 |
| KenShed2 | 7861 | 2328 | 3 | 1602 | 1418.6 | 24/13.2 |
| TanzShed1 | 8092 | 1244 | 3 | 1137 | 1137.8 | 28.7/17.5 |
| TanzShed2 | 2154 | 1097 | 3 | 1428 | 1136.2 | 28.2/17.8 |

Table 2: An example of the statistics used to compare ground rainfall data with satellite products (e.g. ARC2, CHIRP, and CHIRPS) in EthioShed1. The three modes of comparison are compared based on a range of statistical variables (section 3.2). The Point (station) to area grid cell average is computed by comparing individual station to the area grid cell average of each product. Best fit of the last three rows is indicated in bold, best fit of the nine stations are also highlighted.

| Station | ARC2 | | | | | | CHIRP | | | | | | CHIRPS | | | | | |
|---|---|---|---|---|---|---|---|---|---|---|---|---|---|---|---|---|---|---|
| | CC | Bias | Rbias | MAE | RMSE | IA | CC | Bias | Rbias | MAE | RMSE | IA | CC | Bias | Rbias | MAE | RMSE | IA |
| Anger | 0.32 | -0.55 | -14.7 | 4.64 | 10.48 | 0.52 | 0.37 | 0.45 | 11.9 | 4.87 | 9.58 | 0.56 | 0.40 | -0.15 | -4.1 | 4.40 | 9.40 | 0.6 |
| Arijo | 0.29 | -1.25 | -37.6 | 5.02 | 10.14 | 0.53 | 0.23 | 0.12 | 3.4 | 5.94 | 10.29 | 0.49 | 0.37 | -0.42 | -12.5 | 5.08 | 9.33 | 0.61 |
| Bedele | 0.33 | -1.40 | -28.7 | 5.00 | 10.67 | 0.55 | 0.34 | -0.45 | -9.3 | 5.21 | 9.32 | 0.55 | 0.41 | -0.54 | -11.1 | 4.96 | 9.30 | 0.62 |
| Dedesa | 0.29 | -0.71 | -18.0 | 4.77 | 10.53 | 0.51 | 0.28 | 0.13 | 3.4 | 5.16 | 10.49 | 0.50 | 0.34 | -0.23 | -5.8 | 4.81 | 9.92 | 0.55 |
| Gimbi | 0.32 | -1.03 | -23.9 | 5.10 | 10.78 | 0.55 | 0.39 | 0.12 | 2.8 | 5.17 | 9.85 | 0.59 | 0.42 | -0.20 | -4.5 | 4.91 | 9.76 | 0.64 |
| Nekemt | 0.44 | -1.20 | -23.4 | 4.71 | 10.69 | 0.64 | 0.38 | 0.02 | 0.3 | 5.79 | 10.96 | 0.59 | 0.41 | -0.75 | -14.6 | 5.30 | 10.44 | 0.62 |
| Alge | 0.32 | -1.08 | -27.6 | 5.13 | 10.2 | 0.55 | 0.36 | -0.45 | -11.7 | 4.99 | 9.12 | 0.56 | 0.37 | -0.36 | -9.4 | 5.36 | 10.01 | 0.6 |
| Ayira | 0.3 | -1.02 | -26.1 | 5.18 | 10.68 | 0.53 | 0.40 | -0.18 | -4.80 | 4.95 | 8.98 | 0.60 | 0.36 | -0.37 | -9.6 | 5.41 | 10.07 | 0.6 |
| Hurma | 0.31 | -1.01 | -25.7 | 5.20 | 10.44 | 0.54 | 0.38 | -0.56 | -14.5 | 4.59 | 8.86 | 0.54 | 0.37 | -0.60 | -15.8 | 5.15 | 9.59 | 0.6 |
| Average of point – pixel | 0.32 | -1.03 | -25.1 | **4.97** | 10.51 | 0.55 | 0.35 | -0.09 | **-2.06** | 5.19 | **9.72** | 0.55 | **0.38** | **-0.40** | -9.7 | 5.04 | 9.76 | **0.6** |
| Average of point – area grid cells average | 0.37 | -1.1 | -26.8 | **4.82** | 9.59 | 0.53 | 0.37 | -0.21 | **-5.5** | 10.18 | 9.51 | 0.54 | **0.40** | **-0.46** | -11.5 | 4.86 | **9.38** | **0.58** |
| Stations average - area grid cells average | 0.58 | -1.3 | -27.3 | 5.66 | 3.22 | 0.74 | 0.58 | -0.27 | **-5.7** | 5.6 | 3.29 | 0.75 | **0.64** | **-0.59** | -12.0 | **5.38** | **3.08** | **0.79** |

Table 3: Evaluation results of multiple daily rainfall products against field meteorological stations covering the period of 1983–2005 for 21 validation areas of East Africa. For ease of comparison, only selected statistical estimators are given in the table. For individual RCMs) their mean (RCMs) is given here. Best fit is indicated in bold.

| Validation area | ARC2 | | | CHIRPS | | | CHIRP | | | ORH | | | RCMs | | |
|---|---|---|---|---|---|---|---|---|---|---|---|---|---|---|---|
| | CC | Rbias | RMSE | CC | Rbias | RMSE | CC | Rbias | RMSE | CC | Rbias | RMSE | CC | Rbias | RMSE |
| EthioShed1 | 0.59 | -31.2 | 5.64 | **0.64** | **-5.7** | **5.5** | 0.57 | -7.2 | 5.7 | 0.20 | -16.7 | 10.81 | 0.52 | -37.2 | 6.1 |
| EthioShed2 | 0.58 | -27.3 | 5.66 | **0.64** | -12.0 | **5.38** | 0.58 | **-5.7** | 5.6 | 0.18 | 19.8 | 8.6 | 0.43 | 45.8 | 4.8 |
| EthioShed3 | 0.63 | -29.4 | 4.68 | **0.69** | -12.6 | **4.37** | 0.64 | -10.8 | 4.46 | 0.25 | -11.7 | 9.86 | 0.60 | **-4.8** | 4.65 |
| EthioShed4 | 0.59 | -37.4 | **4.95** | **0.61** | 1.9 | 5.34 | 0.49 | 3.9 | 5.35 | 0.12 | -1.5 | 12.14 | 0.30 | 38.7 | 6.38 |
| EthioShed5 | 0.40 | -11.8 | 4.65 | **0.43** | 12.1 | 4.72 | 0.39 | **8.7** | **4.35** | 0.10 | 24.5 | 8.48 | 0.14 | -9.3 | 4.68 |
| EthioShed6 | 0.47 | -42.6 | **3.55** | **0.64** | **5.20** | 3.66 | 0.47 | 5.6 | 3.65 | 0.11 | 16.8 | 7.73 | 0.21 | 60.5 | 6.74 |
| EthioShed7 | 0.55 | -27.8 | 5.27 | **0.70** | **-1.4** | **4.68** | 0.49 | 4.9 | 5.64 | 0.12 | -13.6 | 10.7 | 0.38 | -3.9 | 5.77 |
| EthioShed8 | 0.33 | -22.7 | 7.29 | **0.46** | -2.7 | 6.56 | 0.44 | **-2.0** | **5.60** | 0.11 | -22.4 | 12.0 | 0.37 | 21.8 | 7.28 |
| EthioShed9 | 0.30 | **-7.2** | 5.16 | **0.33** | -9.4 | 4.43 | 0.28 | -26.0 | **4.03** | 0.06 | -22.1 | 7.46 | 0.07 | -39.4 | 4.38 |
| EthioShed10 | 0.59 | -38.2 | **4.36** | **0.60** | **-0.7** | 4.81 | 0.53 | -2.7 | 4.70 | 0.18 | 14.8 | 10.92 | 0.45 | 39.4 | 5.21 |
| EthioShed11 | 0.45 | -38.1 | 4.58 | **0.48** | **0.2** | 4.86 | 0.43 | -0.9 | **4.37** | 0.10 | -10.9 | 7.33 | 0.13 | -51.9 | 4.73 |
| EthioShed12 | **0.42** | -31.3 | **3.75** | 0.35 | 31.5 | 4.15 | 0.32 | 24.3 | 4.0 | 0.1 | 35.1 | 5.58 | 0.07 | **-14.8** | 4.10 |
| EthioShed13 | 0.46 | -38.0 | 5.50 | **0.52** | -14.4 | **5.2** | 0.37 | -15.4 | 5.70 | 0.13 | -13.2 | 9.35 | 0.26 | **12.4** | 5.92 |
| EthioShed14 | 0.40 | -14.3 | 4.76 | **0.41** | **2.0** | 4.78 | 0.35 | -2.0 | **4.53** | 0.10 | 13.7 | 7.85 | 0.12 | -48.7 | 4.75 |
| EthioShed15 | 0.39 | -35.3 | 4.72 | **0.45** | **-11.3** | **4.58** | 0.35 | -15.5 | 4.74 | 0.11 | 97.2 | 8.4 | 0.18 | 37.9 | 5.11 |
| EthioShed16 | 0.29 | -42.3 | 5.06 | **0.35** | 12.2 | 5.72 | 0.29 | **3.9** | **5.04** | 0.12 | 12.7 | 7.89 | 0.16 | 11.2 | 5.17 |
| EthioShed17 | 0.45 | -25.4 | 3.89 | 0.56 | **7.0** | 3.82 | 0.46 | 9.3 | **3.75** | 0.13 | 23.8 | 7.67 | 0.20 | -14.4 | 4.1 |
| KenShed1 | **0.62** | **6.4** | **3.5** | 0.4 | 58.6 | 5.56 | 0.31 | 45 | 5.60 | 0.36 | 21 | 5.65 | 0.1 | 22.1 | 4.72 |
| KenShed2 | **0.72** | -22 | **4.39** | 0.38 | **-4.9** | 8.72 | 0.38 | 24.2 | 6.88 | 0.5 | 22.1 | 7.23 | 0.2 | 67.3 | 7.56 |
| TanzShed1 | 0.3 | -40 | 5.83 | **0.43** | -29 | **5.7** | 0.40 | 30.7 | 6.03 | 0.24 | **13.7** | 7.44 | 0.13 | 38.2 | 6.4 |
| TanzShed2 | 0.3 | **7.2** | 0.23 | **0.44** | 19 | **0.2** | 0.38 | 36.2 | 5.5 | 0.11 | 12 | 0.3 | 0.21 | 22 | 0.21 |

Table 4: Summary of daily rainfall characteristics retrieved from multiple rainfall products and averaged over the validation areas of Ethiopia, Kenya and Tanzania. Values in brackets give the deviation from the observed value (%). The value which comes closest to the observed value is highlighted in bold.

| Rainfall characteristics | Obs. | ARC2 | CHIRP | CHIRPS | ORH | HadGEM2 | MPI | GFDL | RCMs |
|---|---|---|---|---|---|---|---|---|---|
| Number of wet days (days/year) | 189.58 | 162.98 (-15.1) | 351.06 (59.7) | **189.26** (-0.17) | 192.14 (1.34) | 205.08 (7.85) | 243.55 (24.92) | 210.42 (10.42) | 299.36 (44.9) |
| Average duration of wet periods (days) | 5.86 | 4.78 (-20) | 167.96 (186) | **5.13** (-13.4) | 3.02 (-64) | 11.70 (66.5) | 12.17 (69.9) | 9.37 (46) | 21.36 (113.9) |
| Total amount of precipitation (mm/year) | 953.63 | 671.62 (-34.7) | **980.24** (2.75) | 912.0 (-4.5) | 1027.02 (7.41) | 841.73 (-12.5) | 1055.7 (10.2) | 1253.38 (27.2) | 1068.6 (11.4) |
| Average amount of wet periods (mm) | 30.20 | 20.56 (-38) | 498.43 (177) | **25.46** (-17) | 15.64 (-63.5) | 50.12 (49.6) | 55.45 (59) | 59.64 (65.6) | 78.88 (89.3) |
| Average duration of dry periods (days) | 5.37 | 6.01 (11.3) | 1.53 (-111.3) | 4.5 (-17.6) | 2.55 (-71.31) | 6.91 (25.04) | **5.67** (5.4) | 6.55 (19.8) | 3.55 (-41) |
| Average daily precipitation (mm/day) | 5.28 | 4.16 (-23.8) | 2.78 (-62) | 4.88 (-7.7) | **5.4** (2.3) | 3.88 (-31.5) | 4.19 (-22.8) | 5.69 (7.6) | 3.48 (-41) |

Table 5: Statistical evaluation of daily T-max retrieved from climate model and reanalysis-based products against ground observations over the period of 1983–2005 for 21 validation areas of East Africa. For ease of comparison, only selected statistical estimators are given in the table. Best fit is indicated in bold.

| Validation areas | ORH | | | HadGEM2 | | | GFDL | | | MPI | | | RCMs | | |
|---|---|---|---|---|---|---|---|---|---|---|---|---|---|---|---|
| | CC | Rbias | RMSE | CC | Rbias | RMSE | CC | Rbias | RMSE | CC | Rbias | RMSE | CC | Rbias | RMSE |
| EthioShed1 | 0.63 | **3.1** | 2.65 | 0.71 | -3.5 | 2.83 | 0.56 | -3.2 | 3.04 | 0.68 | -6.2 | -6.2 | **0.72** | -4.3 | **2.56** |
| EthioShed2 | **0.63** | **-7.9** | **2.77** | 0.57 | -17.0 | 5.07 | 0.48 | -21 | 5.95 | 0.51 | -19.3 | 5.57 | 0.63 | -19.1 | 5.30 |
| EthioShed3 | 0.71 | **4.1** | **2.30** | 0.77 | -6.7 | 2.72 | 0.39 | -6.8 | 3.24 | 0.73 | -8.9 | 3.09 | **0.78** | -7.5 | 2.53 |
| EthioShed4 | **0.64** | **5.0** | **2.67** | 0.42 | -15.6 | 4.95 | 0.52 | -19.1 | 5.74 | 0.43 | -16.9 | 5.22 | 0.56 | -17.2 | 5.06 |
| EthioShed5 | 0.61 | **2.1** | **2.25** | 0.63 | **-5.6** | 3.12 | 0.46 | -10.3 | 3.98 | 0.62 | 10.1 | 3.83 | **0.65** | -8.7 | 3.23 |
| EthioShed6 | **0.70** | **-3.3** | **1.69** | 0.53 | -11.6 | 3.51 | 0.45 | -19.4 | 4.87 | 0.48 | -15.6 | 4.16 | 0.58 | -15.6 | 3.92 |
| EthioShed7 | 0.63 | **-2.8** | **2.30** | 0.63 | -13.8 | 4.48 | 0.52 | -14.5 | 4.77 | 0.64 | -16.7 | 4.99 | **0.66** | -15.0 | 4.51 |
| EthioShed8 | 0.63 | **1.4** | **2.33** | 0.65 | **-7.7** | 3.31 | 0.56 | -10.8 | 4.10 | 0.65 | -12.4 | 4.11 | **0.69** | -10.3 | 3.50 |
| EthioShed9 | **0.35** | **2.3** | **2.59** | 0.30 | -7.4 | 3.51 | 0.28 | -12.8 | 4.89 | 0.21 | -9.5 | 4.1 | 0.33 | -9.9 | 3.81 |
| EthioShed10 | **0.51** | 17.8 | 4.40 | 0.45 | **-0.5** | 2.70 | 0.34 | -4.1 | 2.90 | 0.39 | -2.9 | 2.72 | 0.50 | -2.5 | **2.27** |
| EthioShed11 | 0.52 | 1.6 | 2.4 | 0.54 | 3.0 | 2.78 | 0.45 | -0.9 | 3.1 | 0.56 | -1.6 | 2.84 | **0.60** | **0.2** | **2.34** |
| EthioShed12 | 0.42 | **-1.2** | **2.23** | 0.43 | -5.3 | 2.96 | 0.16 | -6 | 3.6 | 0.44 | -5.7 | 3.10 | **0.50** | -4.6 | 2.50 |
| EthioShed13 | **0.4** | 17.5 | 5.77 | 0.33 | **-5.0** | 3.97 | 0.29 | -7.5 | 4.6 | 0.32 | -6.4 | 4.15 | 0.37 | -6.3 | **3.90** |
| EthioShed14 | **0.51** | **0.1** | **2.72** | 0.43 | -11.4 | 4.8 | 0.41 | -15.8 | 6.17 | 0.38 | -13.1 | 5.28 | 0.47 | -13.5 | 5.20 |
| EthioShed15 | 0.22 | **3.0** | **3.1** | 0.26 | -6.3 | 3.9 | 0.3 | -9.8 | 3.9 | 0.14 | -10.2 | 4.5 | 0.27 | -8.8 | 3.73 |
| EthioShed16 | **0.4** | **-3.1** | **3.45** | 0.23 | -12.7 | 5.1 | 0.25 | -19.4 | 6.4 | 0.24 | -15.2 | 5.45 | 0.30 | -15.8 | 5.43 |
| EthioShed17 | **0.62** | 5.2 | **2.3** | 0.58 | **-3.6** | 2.92 | 0.45 | -7.8 | 3.2 | 0.53 | -7.6 | 3.31 | 0.61 | -6.4 | 2.66 |
| KenShed1 | **0.59** | 9.6 | 3.2 | 0.39 | 4.6 | 3.03 | 0.37 | **0.9** | 2.94 | 0.34 | 2.3 | 2.85 | 0.46 | 2.6 | **2.46** |
| KenShed2 | **0.65** | -7.3 | **2.62** | 0.48 | **-7.1** | 3.02 | 0.4 | -17.2 | 4.97 | 0.41 | -11.8 | 3.83 | 0.53 | -12.1 | 3.62 |
| TanzShed1 | **0.66** | **-5.5** | **3.11** | 0.56 | -9.2 | 4.16 | 0.39 | -11.4 | 4.82 | 0.48 | -11.1 | 4.58 | 0.58 | -10.6 | 4.20 |
| TanzShed2 | **0.48** | **-4.1** | **2.8** | 0.35 | -14 | 4.9 | 0.22 | -13.9 | 5.03 | 0.35 | 16.4 | 5.4 | 0.40 | -14.8 | 4.90 |

Table 6: Statistical evaluation of daily T-min retrieved from climate model and reanalysis-based products against ground observations over the period of 1983–2005 for 21 validation areas of East Africa. For ease of comparison, only selected statistical estimators are given in the table. Best fit is indicated in bold.

| Validation areas | ORH | | | HadGEM2 | | | GFDL | | | MPI | | | RCMs | | |
|---|---|---|---|---|---|---|---|---|---|---|---|---|---|---|---|
| | CC | Rbias | RMSE | CC | Rbias | RMSE | CC | Rbias | RMSE | CC | Rbias | RMSE | CC | Rbias | RMSE |
| EthioShed1 | **0.54** | 7.1 | **1.76** | 0.45 | 12.8 | 2.55 | 0.37 | **3.7** | 2.44 | 0.4 | 10.4 | 2.32 | 0.51 | 9.0 | 1.97 |
| EthioShed2 | **0.77** | -11.3 | **2.11** | 0.59 | **-6** | 2.2 | 0.54 | -18.7 | 3.27 | 0.58 | -7.1 | 2.28 | 0.67 | -10.7 | 2.14 |
| EthioShed3 | **0.65** | 6.8 | **2.12** | 0.55 | 15.5 | 2.71 | 0.52 | **4.9** | 2.31 | 0.51 | 13.3 | 2.58 | 0.62 | 11.2 | 2.20 |
| EthioShed4 | **0.76** | 12.1 | **2.50** | 0.60 | -12.3 | 3.01 | 0.45 | -26.8 | 4.71 | 0.61 | **-11.6** | 3.07 | 0.62 | -16.9 | 3.28 |
| EthioShed5 | **0.45** | -10.9 | 2.65 | 0.28 | 6.9 | 2.28 | 0.31 | -3.9 | 2.29 | 0.22 | 3.9 | 2.15 | 0.36 | **2.3** | **1.78** |
| EthioShed6 | **0.69** | -10.4 | **1.9** | 0.53 | 16.5 | 2.38 | 0.47 | **6.7** | 2.38 | 0.49 | 15.9 | 2.42 | 0.61 | 13.1 | 2.03 |
| EthioShed7 | **0.63** | -7.6 | **2.01** | 0.23 | 9.4 | 2.60 | 0.26 | **-0.5** | 2.87 | 0.29 | 6.4 | 2.30 | 0.35 | 5.1 | 2.03 |
| EthioShed8 | **0.33** | **-0.1** | **1.65** | 0.24 | 16.7 | 3.08 | 0.21 | 8.6 | 2.78 | 0.15 | 12 | 2.6 | 0.27 | 12.4 | 2.46 |
| EthioShed9 | **0.68** | 7.5 | **2.84** | 0.64 | **-2.1** | 2.87 | 0.59 | -8.0 | 3.82 | 0.58 | -1.8 | 3.13 | 0.65 | -4.0 | 2.91 |
| EthioShed10 | **0.67** | 16.2 | 2.58 | 0.50 | 9.4 | 2.46 | 0.38 | **-3.1** | 2.66 | 0.50 | 8.8 | 2.51 | 0.54 | 5.0 | **2.13** |
| EthioShed11 | **0.36** | -17.2 | 3.22 | 0.18 | 17.8 | 3.48 | 0.24 | **13.8** | 3.20 | 0.16 | 15.8 | 3.28 | 0.27 | 15.8 | **3.10** |
| EthioShed12 | **0.46** | -6.6 | 2.47 | 0.41 | -3.8 | 2.40 | 0.34 | -6.6 | 2.9 | 0.39 | **-1.9** | 2.26 | 0.45 | -4.1 | **2.21** |
| EthioShed13 | **0.57** | 31.2 | 4.77 | 0.54 | 2.7 | 2.76 | 0.46 | -8.9 | 3.42 | 0.54 | 2.3 | 2.86 | 0.56 | **-1.3** | **2.67** |
| EthioShed14 | **0.72** | **4.6** | **2.68** | 0.61 | -5.6 | 3.27 | 0.55 | -16.3 | 4.62 | 0.59 | -5.5 | 3.32 | 0.63 | -9.2 | 3.40 |
| EthioShed15 | **0.62** | -1.8 | **2.16** | 0.41 | 9.8 | 2.61 | 0.44 | **0.5** | 2.41 | 0.36 | 6.7 | 2.54 | 0.51 | 5.7 | 2.19 |
| EthioShed16 | **0.50** | -8.2 | **3.45** | 0.42 | -7.7 | 3.7 | 0.31 | -23.1 | 4.98 | 0.42 | **-7.1** | 3.66 | 0.44 | -12.7 | 3.77 |
| EthioShed17 | **0.61** | **7.1** | **2.17** | 0.43 | 19.7 | 2.96 | 0.44 | 9.0 | 2.44 | 0.36 | 17.1 | 2.87 | 0.53 | 15.3 | 2.46 |
| KenShed1 | **0.52** | 14.8 | 2.66 | 0.31 | 9.0 | 2.44 | 0.17 | **3.2** | 2.46 | 0.3 | 10.4 | 2.52 | 0.34 | 7.5 | **2.18** |
| KenShed2 | **0.40** | -21.3 | 3.26 | 0.25 | -18.1 | 3.15 | 0.25 | -24.6 | 4.0 | 0.32 | **-16.1** | **2.88** | 0.35 | -19.6 | 3.15 |
| TanzShed1 | **0.53** | **-12.0** | **3.12** | 0.44 | -15 | 3.69 | 0.38 | -18.9 | 4.23 | 0.44 | -15.3 | 3.71 | 0.5 | -16.2 | 3.72 |
| TanzShed2 | 0.51 | **-16.2** | **3.87** | 0.58 | -17.3 | 3.9 | 0.46 | -16.8 | 4.04 | 0.58 | -18.5 | 4.16 | **0.61** | -17.5 | 3.93 |

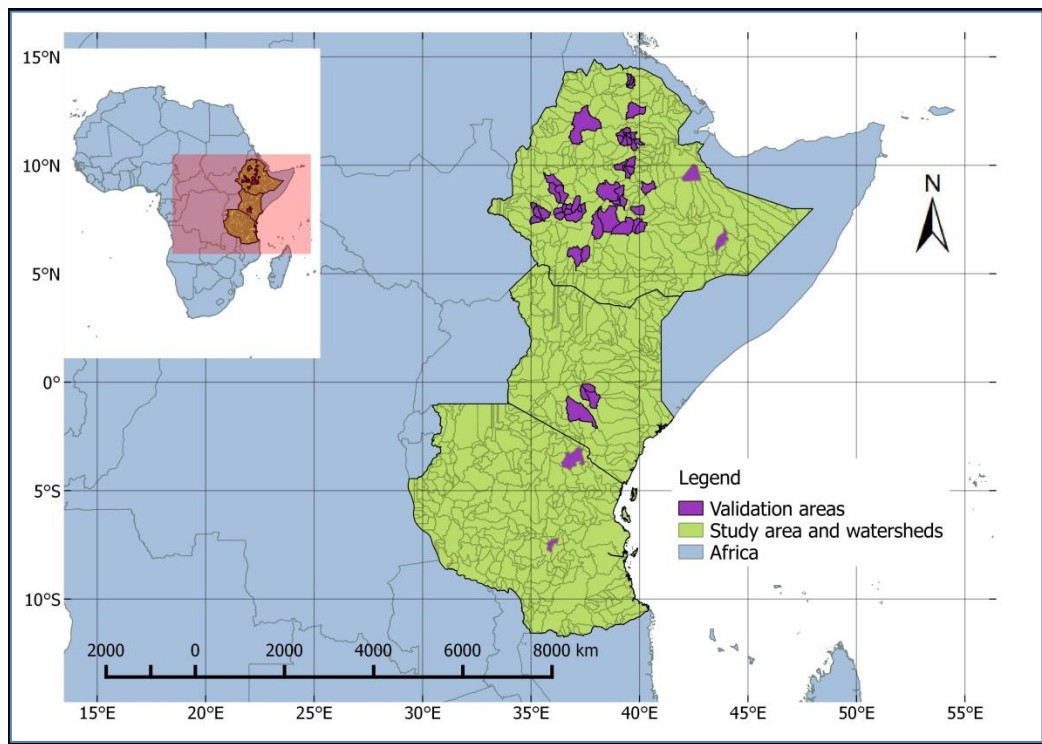

Figure 1: Map of Africa and study regions (Ethiopia, Kenya, and Tanzania) with data validation areas (EthioShed1–17, KenShed1&2, and TanzShed1&2). The basins are retrieved from the WaterBase global data portal (http://www.waterbase.org/).

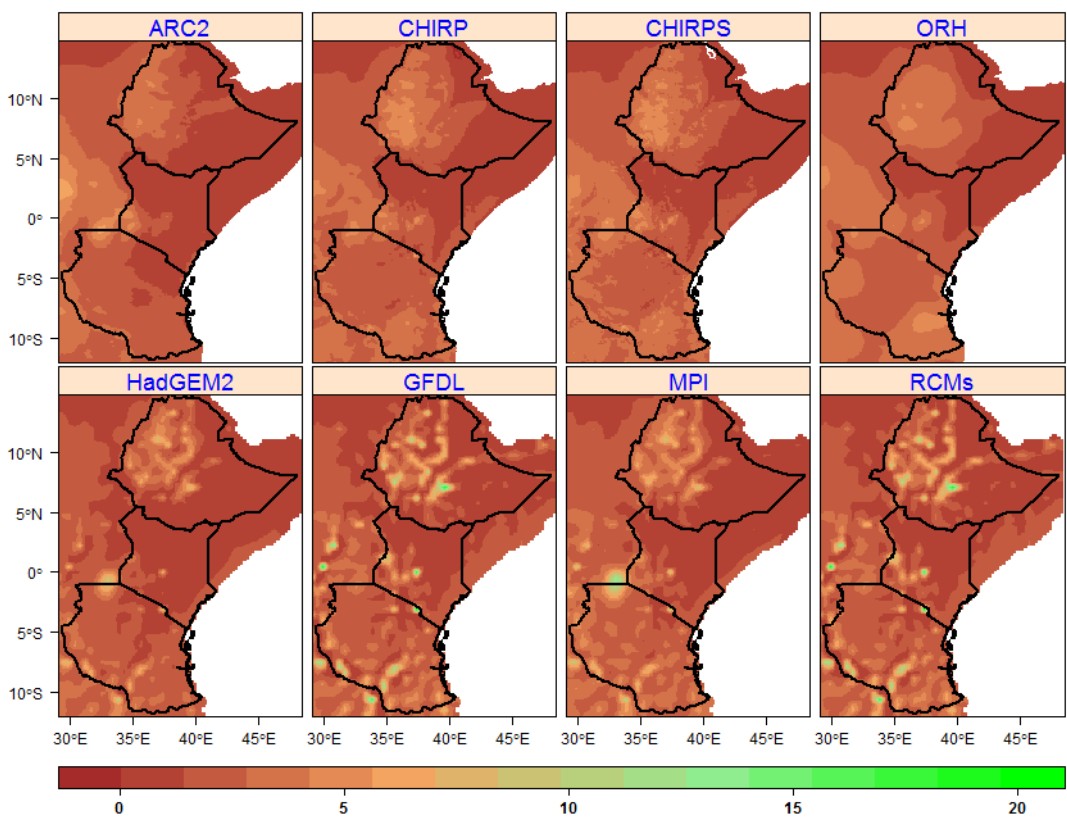

Figure 2: Average daily rainfall (mm day[-1]) maps of East Africa retrieved from ARC2, CHIRP, CHIRPS, ORH, RCM, and RCMs for the study period 1983–2005. All the maps are given in a 0.05° spatial resolution.

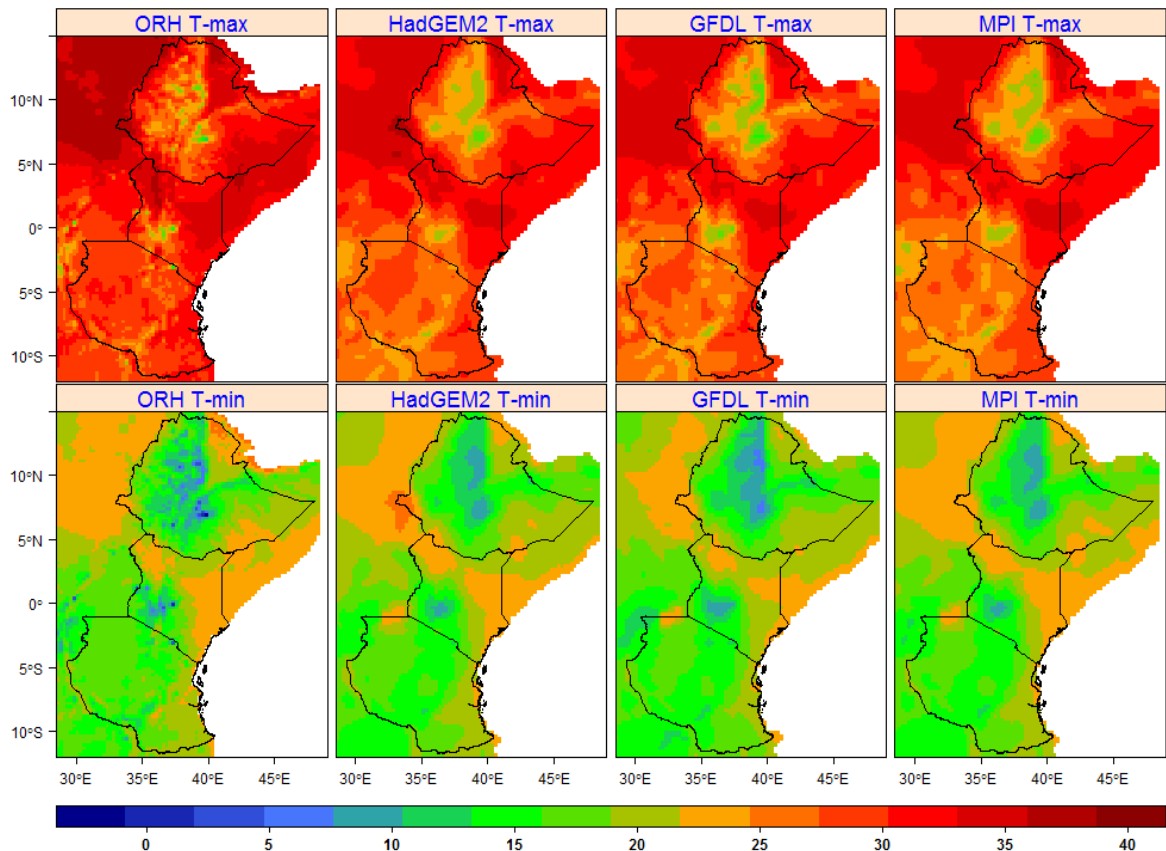

Figure 3: Maps of average daily T-max and T-min (°C) for East Africa generated from ORH and RCMs for the study period 1983–2005. All the maps are given in a 0.1° spatial resolution.

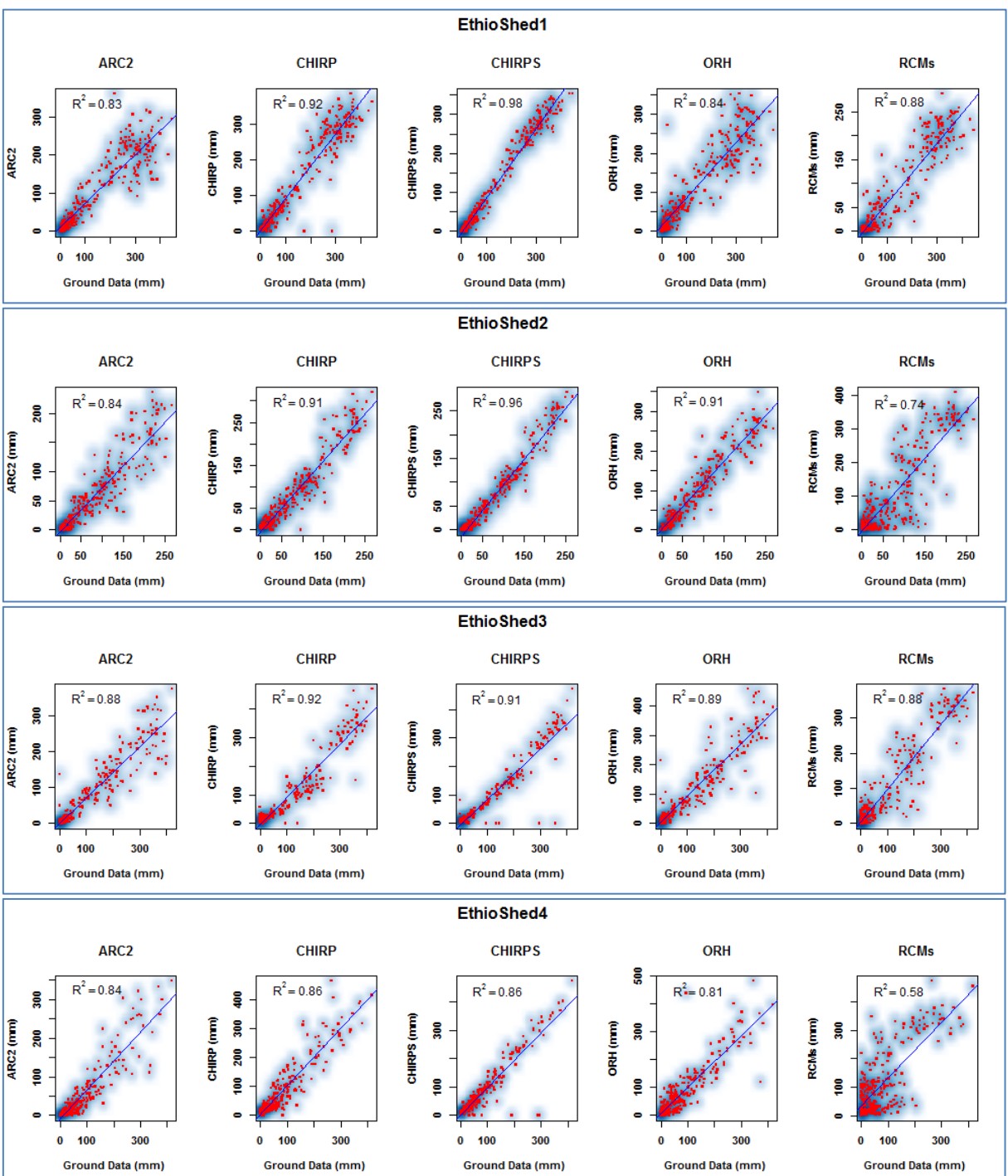

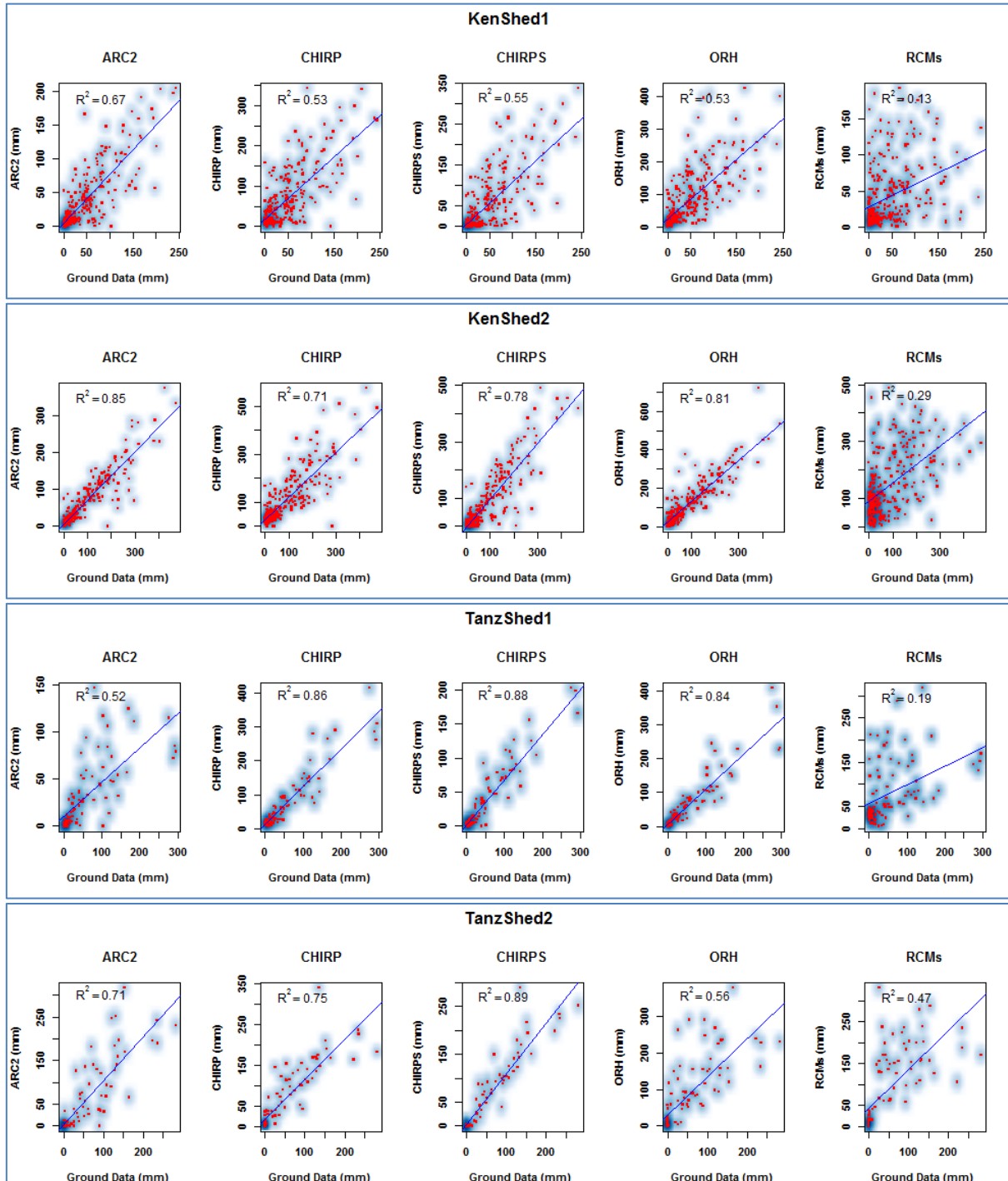

Figure 4: Scatter plots of monthly rainfall for ARC2, CHIRP, CHIRPS, ORH, and RCMs for eight validation areas covering the period of 1983–2005 and aggregated from daily data. Shaded area displays the data density around the regression line.

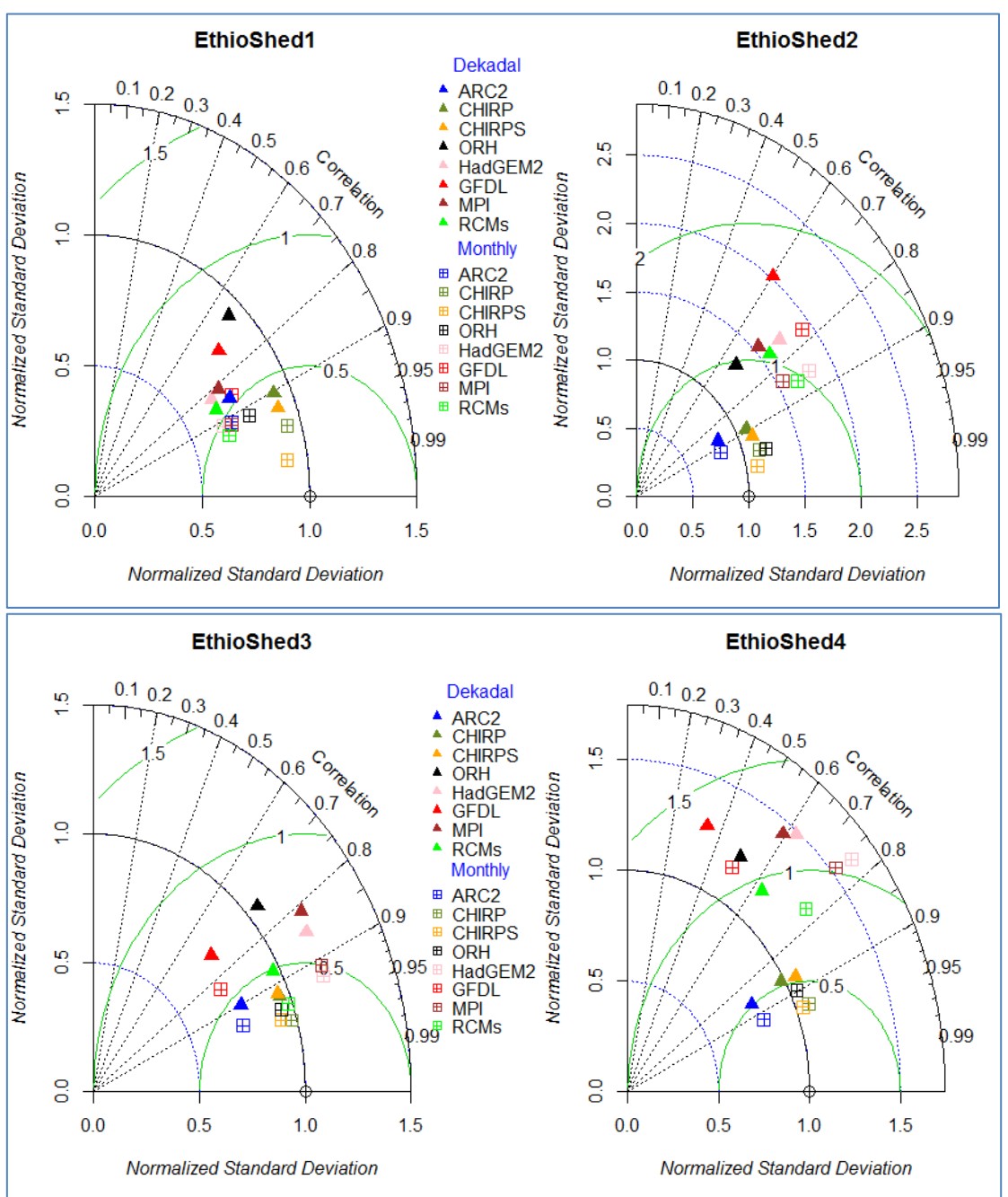

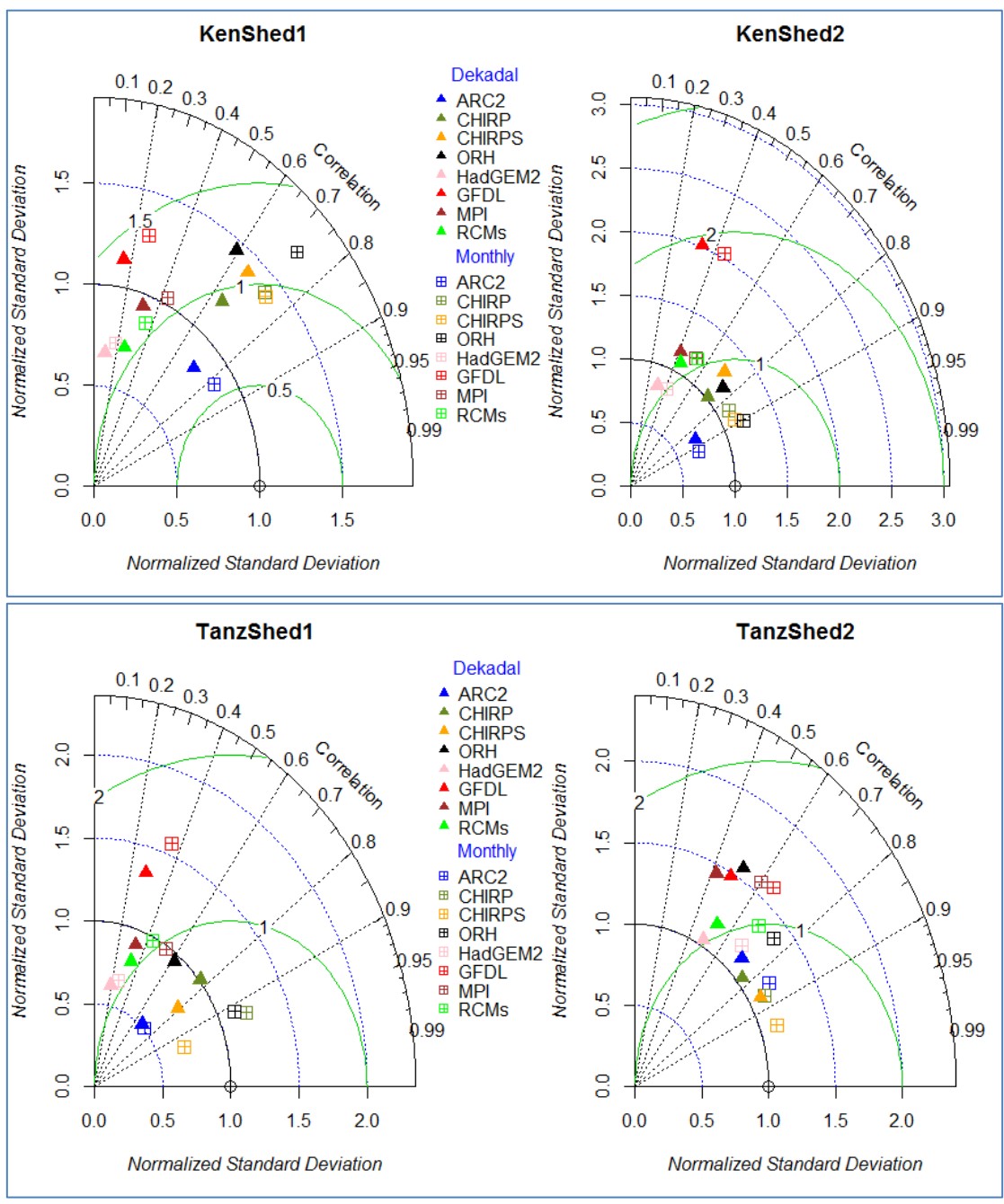

Figure 5: Taylor diagram displaying the agreement between ground observation and synthesized dekadal and monthly rainfall over eight validation areas of Ethiopia, Kenya, and Tanzania covering the period of 1983– 2005.

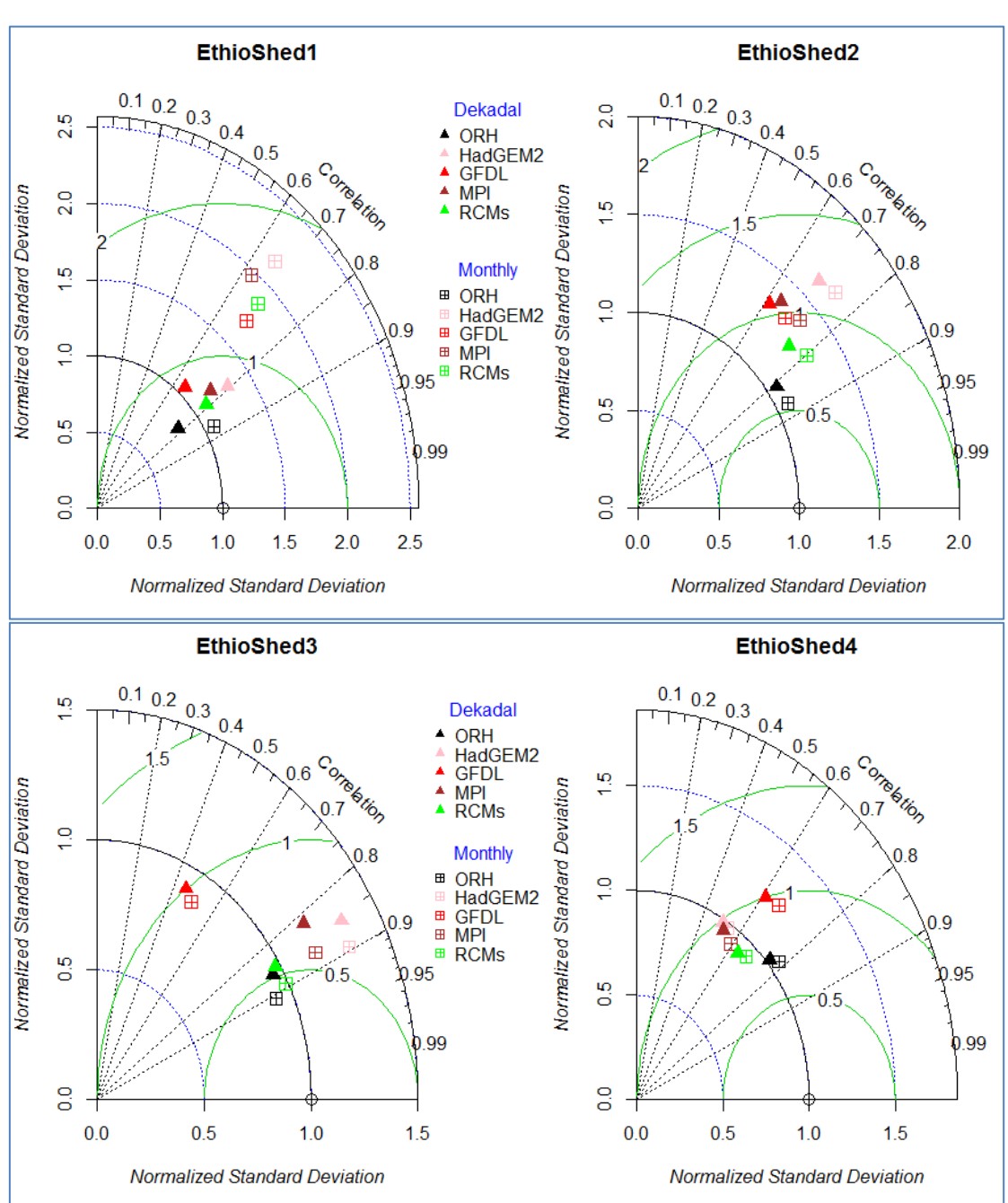

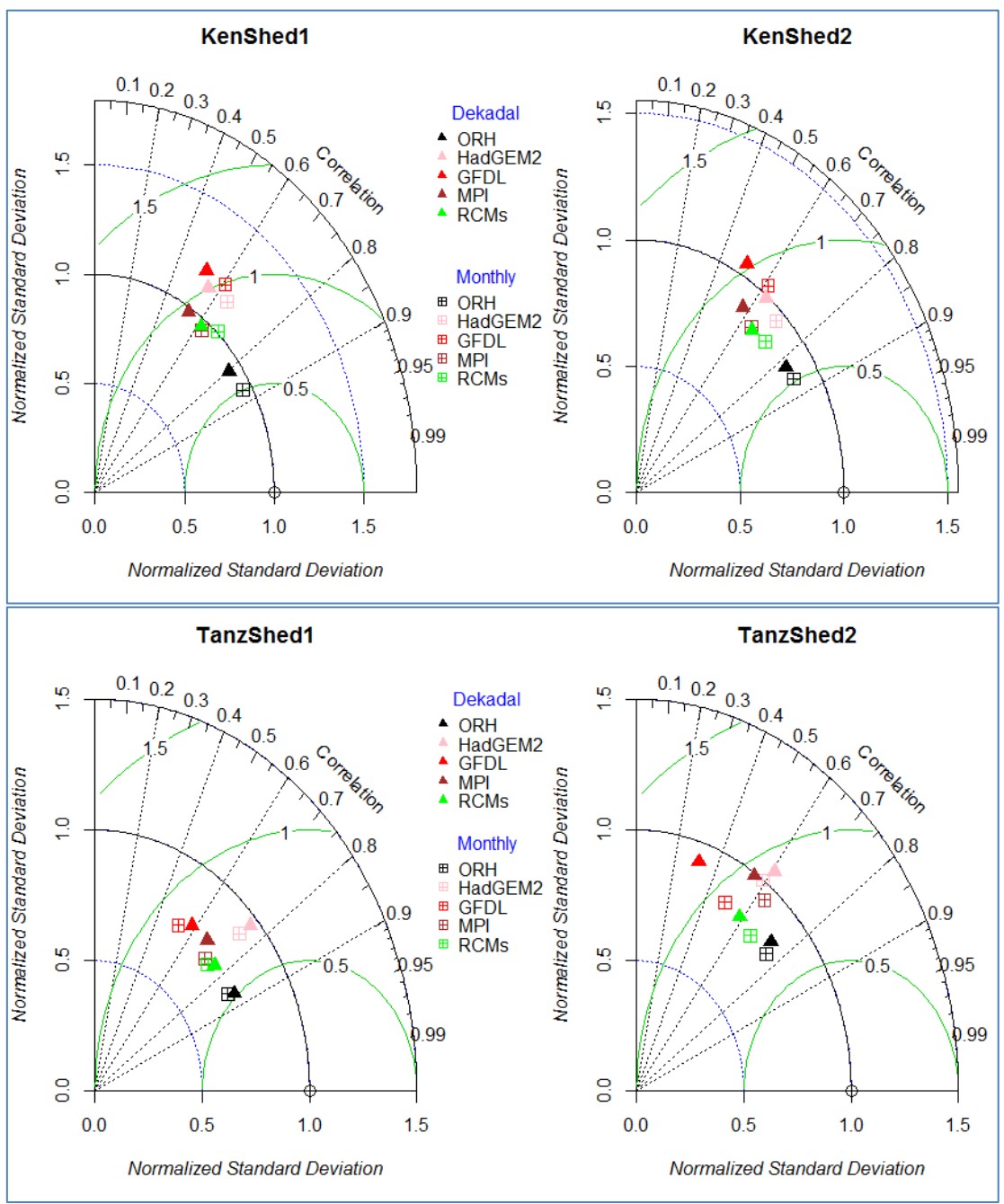

Figure 6: Taylor diagram displaying the agreement between ground observation and synthesized dekadal and monthly T-max over the eight validation areas of Ethiopia, Kenya, and Tanzania covering the period of 1983–2005.

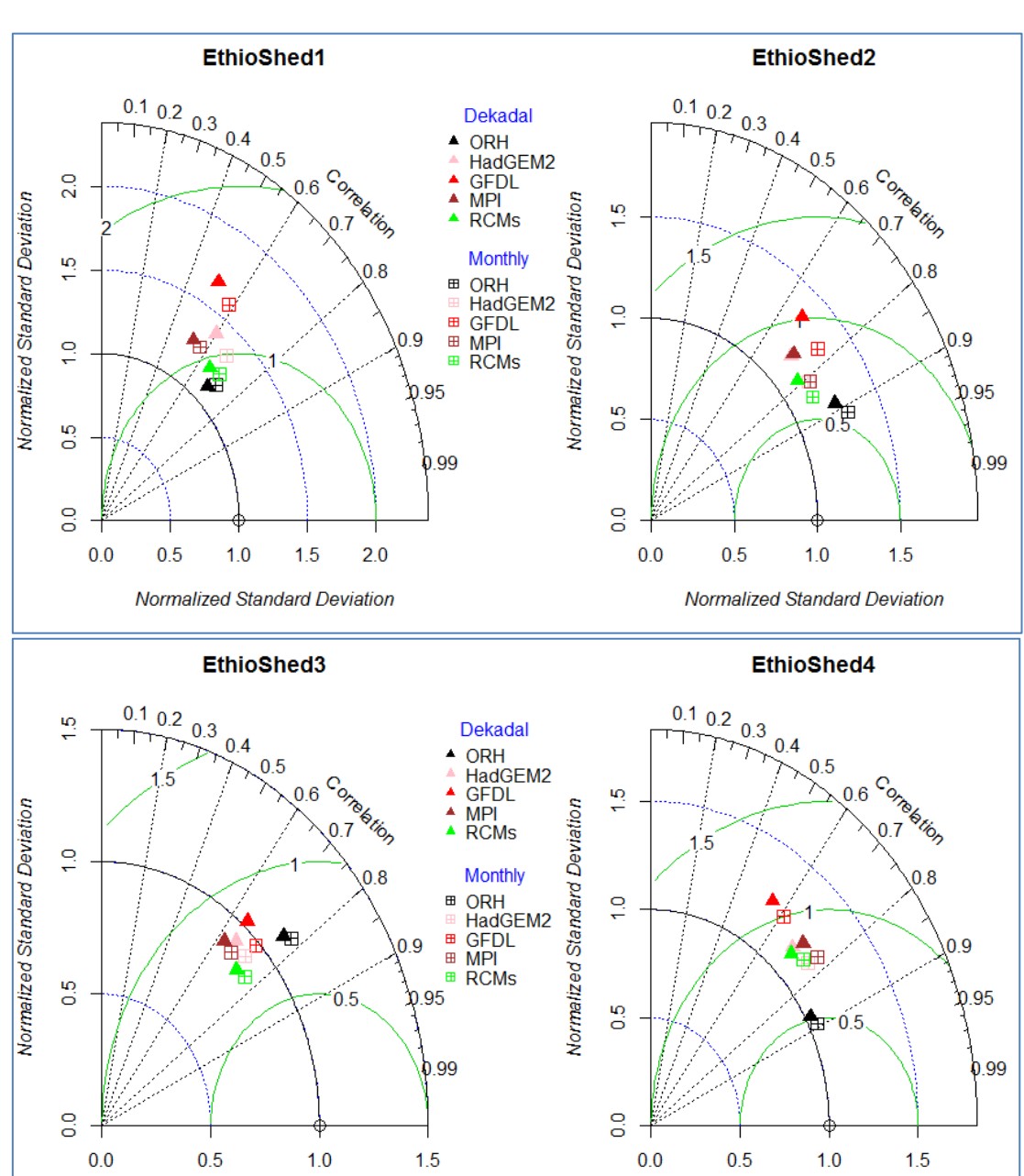

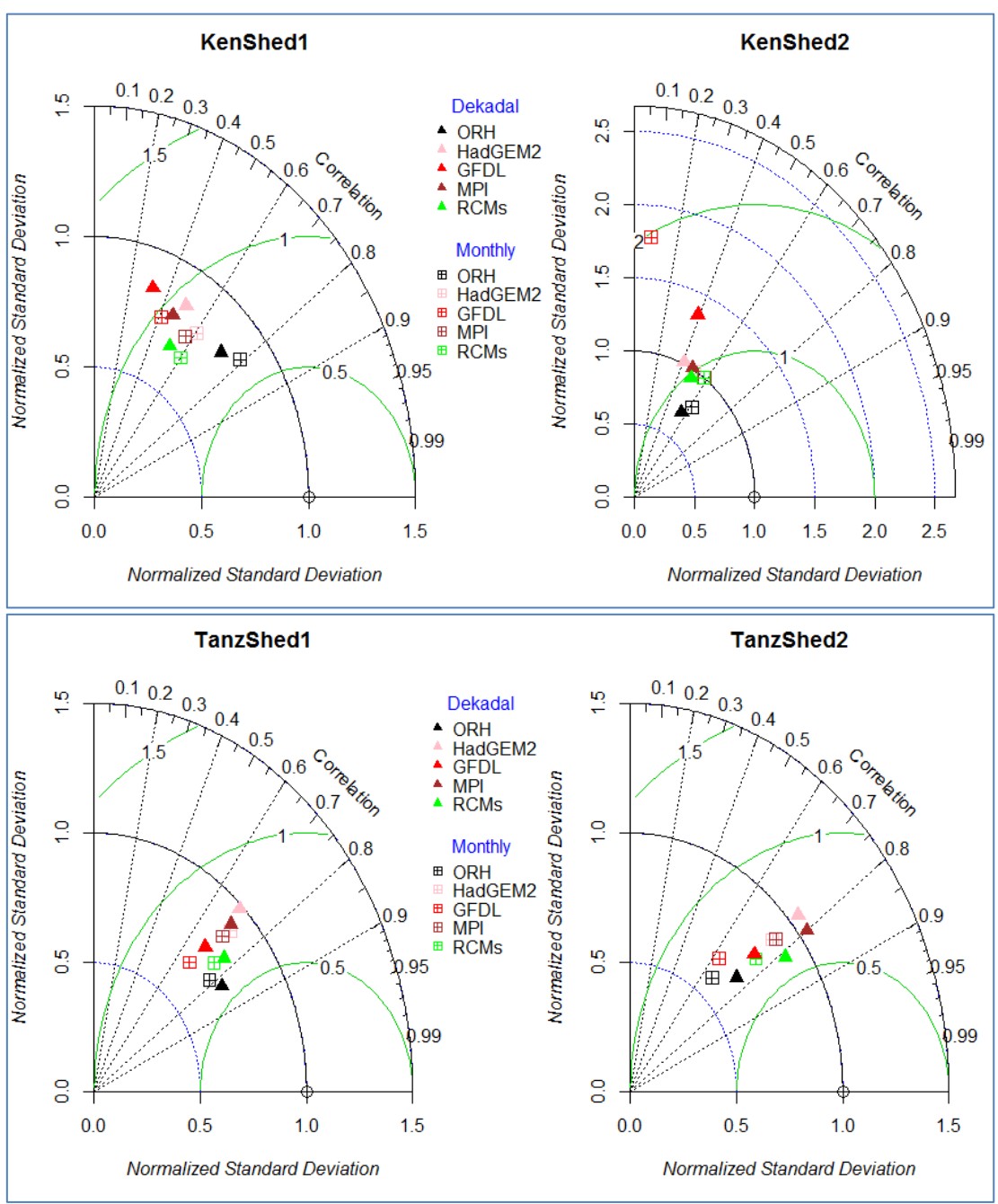

Figure 7: Taylor diagram displaying the agreement between ground observation and synthesized dekadal and monthly T-min over eight validation areas of Ethiopia, Kenya, and Tanzania covering the period of 1983– 2005.

**Acknowledgements**

We would like to thank the National Meteorological Agency (NMA) of Ethiopia for providing adequate data for the study and the World Meteorological Organization (WMO) representative for Eastern and Southern Africa for their kind support in data collection. We would also like to thank the Graduate Academy of Technische Universität Dresden (TU Dresden) for its financial support during the study period.

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
