# Peer review of "Evaluation of Multiple Climate Data Sources for Managing Environmental Resources in East Africa"

_Hydrology and Earth System Sciences, 2017_

## Referee Comment (RC1) · Anonymous Referee #1 · 11 Dec 2017

General Comments This paper compared different rainfall datasets over East Africa that have 30+ year record, to station data in Ethiopia, Kenya and Tanzania. The authors found that the CHIRPS rainfall and the ORH Tmin/Tmax are the best products to use for long-term climate studies (trend, variability, and extreme indices) and input for climate or hydrological models.

While I think this paper is a good start to necessary analysis of daily rainfall products, I have concerns with the lack of independent station data and the narrow scope of the research (results are only regionally relevant, not very generalizable). I think there are ways to work around this problem of data validation in sparse-regions but the authors

would need to reframe the paper and consider how to address the greater challenge of evaluating the quality of satellite rainfall in data sparse regions. The authors also need to be more transparent/detailed about their methods (metrics and data sets).

Specific Comments (Major)

The focus on daily rainfall is useful/novel as the authors state that this has not been done before. In general this is a regional study that has limited applicability to studies beyond Ethiopia, Tanzania and Kenya, and to the extent that this is generalizable in country in questionable judging from Figure. 1. Not to say that research can't be done in data sparse regions but it has to be framed appropriately, and think that the authors could improve in this respect. In fact, "how to evaluate rainfall and temperature in a data sparse region?" is a good question, although i don't think comparing to a handful of stations that are not independent is necessarily the answer.

Major concern is the use of the EMA and GSOD data for evaluation and the conclusion that CHIRPS is the best performing product. I do think that CHIRPS is a very good product (from prior monthly/season scale evaluations and performance in hydrologic models & compared to other remotely sensed data), and it does need to be more carefully evaluated at the daily timestep.

The station data that goes into the different rainfall products needs to be described in the methods/data. In addition to the discussion. CHIRPS includes stations from several sources including GTS and GSOD, ARC includes GTS. Please include information on what stations the other products blend in. The authors indicate that GSOD is only used in the CHIRPS monthly totals making the "dependency rather weak and indirect" Seems to me incorporating GSOD would contribute to the strong monthly correlations in Figure 4. From my interpretation of Funk et al. (2015) the GSOD data is included for pentad-totals as well. You may want to ask the data producers to clarify (and then include that information in the data/methods here. I* think* Ethiopia NMA stations are included in CHIRPS. Check with the date providers Funk et al. 2015 says: "Additional observations have been provided by national meteorological agencies, primarily in Mexico, Central America, South America, and sub-Saharan Africa" apparently ORH also uses GSOD "assimilating quality-controlled and gap-filled Global Summary of the Day (GSOD) in situ measurement" ...what spatial interpolation method do they use?

Evaluation of daily rainfall for trend/variability/extremes/hydro model input is a worth-while goal. Also not sure if the authors accomplished this given that i have questions about their metrics. I understand that you are comparing to stations but... Daily rainfall intensity: intensity is depth per unit time. How are you getting this when you just have daily totals? And then how does the "intensity" metric differ from what you describe as daily totals? Please include your definition of intensity. Number of wet/dry days: is this just a count that does/does not match the stations? Or are you using something like probability of detection and false alarm rate? These metrics need to be defined in the methods. I can't really tell what you did to come up with the results on page 13. Not obvious what "point to area-grid-cell" average means. I gather that its the average over the polygons shown in Figure 1, but this needs more explanation in the methods. Where do these polygons come from? Is there a reason why this level of basin was used to define the watersheds for a country? Since you're not comparing to hydrological/streamflow data why not just average from 0.05 to 0.25 degree - essentially producing the same results as what you discuss with the coarser CHIRPS data?

Specific comments (Minor)

Additional information on how the data is produced should help explain your results (e.g. why is point to area-average best, does this have to do with the interpolation schemes that ARC and CHIRPS and the other product use? Do CHIRPS results improve at 0.25deg because that is its original resolution, before being downcaled to 0.05deg with the CHPclim? This kind information will be useful for the other products as well.

Is only a historic record needed for env. Management? ORH isn't updated regularly

(2012?) This should be clear in the paper, pls include in methods. Meanwhile, ARC & CHIRPS are updated regularly. It will help contextualize the metrics if you discuss the products strengths and weaknesses more with some example of an environmental management application that they might be used for. I am sure there are some that would benefit from ORH long record, or ARC's 1-day latency. If you are including OHR why not include what they use routinely in the Africa Flood and Drought monitor? 3B42RT...how does blending datasets impact the application to environmental management? With respect to hydrologic modeling GLDAS (Rodell et al. 2004) uses ORH/Princeton+other, Africa flood and drought monitor (Sheffield et al. 2014) uses ORH/TRMM-RT, FLDAS (McNally et al. 2017) uses CHIRPS. How does all this relate to the climate models? > intro was vague and too focused on data scarcity - we have lots of data (models, remote sensing, some in situ)...just not lots of dense rainfall stations. > there are lots of datasets to get temperature (e.g. MERRA-2, CFS-R). Why weren't these included?

Technical corrections > fix citation (also 2017): Kimani, M., Hoedjes, J. and Su, Z.: Uncertainty Assessments of Satellite Derived Rainfall Products, , 15 doi:10.20944/preprints201611.0019.v1, 2016. Don't cite the pre-print use this one: Kimani, Margaret Wambui, Joost CB Hoedjes, and Zhongbo Su. "An Assessment of Satellite-Derived Rainfall Products Relative to Ground Observations over East Africa." Remote Sensing 9.5 (2017): 430.

Typo RFE pg 6...its RFE Rainfall Estimation Version 2 (REF 2.0) (Novella et al., 2013)

---

## Referee Comment (RC2) · Anonymous Referee #2 · 20 Dec 2017

General comments This paper is specifically validating the quality of three climatic variables coming from different satellites data-streams and models using scientifically proven quality validation methodologies. The three include rainfall, maximum temperature and minimum temperature. Being a research that has been done for the first time that I know of, the paper unravels the different quality of each of these datasets and with evidence provide great knowledge of which is the best among the 6 datasets for each variable. If further validate the same dataset with observed rainfall and satellite from weather stations. Though not conclusive, through this research, one can relate that the CHIRPS dataset is better for rainfall analysis in specific areas which have complex topography with a case study in three East African countries. While the ORH

dataset works best for the Tmax/ Tmin variable. The paper specifically highlight the methods used and why and how each one is best. East Africa being a complex region of climate analysis. The paper seems to have limited itself to specific sites which might not fully represent the entire region. Despite having fewer observed data the sample areas of interest might limit the imagination of the complexity of the region. CHIRPS products seemed to work well in some areas while at the same time came in second in other areas. The author should try and indicate by what percentage in all the analysis done was CHIRPS top and if the percentage is worth representing the region as the best dataset. Specific comments In page 9 of the document the author mentions that "The quality of selected stations was checked and extremely high rainfall records during dry seasons were excluded." Through this statement it is not clear what is considered as a dry season and the reason for exclusion of such rainfall dataset remains hanging. Also in consideration of the same, extreme event such as flash floods may be recorded in a single days' rainfall. A few questions to be asked are; Could the x,y decimal places affect the location of a given station ending up reporting a value for a wrong location? For example a station reading of 36.123456, -1.123456 might fall at a different location compared to a reading of 36.123, -1.123. In this reference were the station locations validated? From this paper it is also not clear what the following terms refer to; Wet days, duration of wet days and average amount of wet periods this might be confusing since they all are represented by one unit which is days. For example, when we talk about wet days we say 10 days. If we talk about duration of wet days do we still say 10 days? The same applies to the average amount of wet periods. From the paper it is very clear that the author highlights CHIRPS as the best rainfall product while ORH as the best temperature product. CHIRPS comes out better than the rest based on the characteristics described by the author in page 14 but the author has not conclusively stated by how much is CHIRPS better than all this other products if you compare all the statistical analysis done. The Author has only highlighted that "In general, the observed rainfall characteristics 15 are well captured by CHIRPS compared to CHIRP, ARC2, ORH, RCM, and RCMs." While at the same time pointed out areas that

ARC2 has performed better that CHIRPS and CHIRP. Regarding the above, in some instances such as EthioShed4 the CHIRP and CHIRPS have equal R squared while in some areas ARC2 came on top. Through the analysis of all the Sheds analyzed what percentage of CHIRPS compared to the rest of the datasets was better. Still in line with that there are some areas where all the R squared were between 0.13 and 0.55, is it possible to elaborate on why such cases occur? Is it the methodology used to model the datasets that limits the correlation with the station data? Another question of concern is what explains the equal value for CHIRP and CHIRPS as portrayed in EthioShed4? In the introduction the paper highlights CHIRPS as a dataset that has both station and satellite data in it. Might this explain the high correlation? Are the same stations in CHIRPS used to validate the CHIRPS product? In conclusion to the specific comments. The paper is very clear on how the validation is done. However, more can be done to ensure that these products are regarded as the best products as indicated by the author. The paper currently is validating the products for areas with low observed dataset. Perhaps, the author can use historical analysis as a means of validation too. Also, an elaborate point of validation would be to highlight how the non-blended datasets such as CHIRP is performing compared to observed station data in regions that have well established network of weather stations such as the developed countries. Then further, validating the CHIRP against the CHIRPS. This will basically ensure less redundancy.

Technical comments

In page 7, the Dekadal should come after pentadal since the former represents 10 days and the later represents 5 days. In page 13, 17 is numerical while three and one are text – you might want to use either for all. In Page 20, it is indicated that "The products are available with higher spatial and temporal resolution and for longer periods." – doesn't longer periods mean the same as temporal resolution?

[Figure]

558, 2017.

---

## Author Comment (AC1) · 20 Jan 2018

We thank the reviewer for reviewing our manuscript and providing his/her valuable feedbacks. We have now addressed all of his/her comments and discussed them in the following. The comments were very helpful to identify some unclear issues with regard to the scope, methodologies and conclusions of the paper. We have revised the manuscript to resolve these issues and make our approach and conclusions more clear-cut. Thanks to the reviewer's feedback, the paper is now much improved.

General Comments
- This paper compared different rainfall datasets over East Africa that have 30+ year record, to station data in Ethiopia, Kenya and Tanzania. The authors found that the CHIRPS rainfall and the ORH Tmin/Tmax are the best products to use for long-term climate studies (trend, variability, and extreme indices) and input for climate or hydrological models. While I think this paper is a good start to necessary analysis of daily rainfall products, I have concerns with the lack of independent station data and the narrow scope of the research (results are only regionally relevant, not very generalizable). I think there are ways to work around this problem of data validation in sparse-regions but the authors would need to reframe the paper and consider how to address the greater challenge of evaluating the quality of satellite rainfall in data sparse regions. The authors also need to be more transparent/detailed about their methods (metrics and data sets).

Authors´ response:
- We appreciate acknowledgement that our paper provides a good start for analysing rainfall products. We would rather propose to speak of scarcity of station data, not a general lack of independent station data. We are convinced that this kind of validation of climate data products can only be done at a regional level and indeed we feel that it is impossible to do this at a significantly larger scale than provided here (3 countries in East Africa). We rephrased respective sections (abstract, lines 6-8), introduction (page 5, lines 18-25) and summary and conclusion (page 20, lines 6-12) to make this point clearer. In fact, earlier studies with a similar goal of validating climate data (e.g., CHIRPS) products did so based on much more limited data basis in terms of temporal resolution or spatial dimension (e.g., *Duan et al., (2016) and Dembélé and Zwart (2016)*). We also think that our general approach is well generalizable and can and should be used for validating these climate data products in other areas as well.
  We added some explanations in the methodology part (section 3.2, page 9, lines 19-21) to be more transparent and clear.

Specific Comments (Major)
- The focus on daily rainfall is useful/novel as the authors state that this has not been done before. In general this is a regional study that has limited applicability to studies beyond Ethiopia, Tanzania and Kenya, and to the extent that this is generalizable in country in questionable judging from Figure. 1. Not to say that research can't be done in data sparse regions but it has to be framed appropriately, and think that the authors could improve in this respect.

Authors´ response:
- Thanks for pointing out the novelty of our study. It is true that the study is a regional study with a main focus in Ethiopia, Kenya and Tanzania. Following this we will modify sections of the paper (e.g., abstract (lines 6-8), introduction (page 5, lines 18-25) and summary and conclusion (page 20, lines 6-12)) to emphasize the regional focus – while pointing out that the approach is applicable in other regions, too.

Concerning framing we not only revised sections explaining the regional focus, but also improved the emphasis on the temporal dimension (daily resolution) and the different modes of data validation, going beyond the typical point to pixel comparison (see below).

- In fact, "how to evaluate rainfall and temperature in a data sparse region?" is a good question, although i don't think comparing to a handful of stations that are not independent is necessarily the answer. Major concern is the use of the EMA and GSOD data for evaluation and the conclusion that CHIRPS is the best performing product.

Authors´ response:

- We felt the same and this is why we posed the question – in this case for East Africa. Concerning the number of stations we did our best to get data from more than just a handful of stations: At least for Ethiopia the data set we used represents the most comprehensive to date and the most comprehensive possible (based on a quality controlled dataset from the National Meteorological Agency of Ethiopia). For Kenya and Tanzania data availability is admittedly thinner, partly due to quite restrictive data sharing policies (as explained in the paper), but the available data in our view do allow making the case that we validated the considered data sources at a regional scale. Adding data from two more countries provides additional insights about the accuracy of these data products. The concern with regard to the independency of data is taken up below. Indeed, the inclusion of station data in CHIRPS may raise such concerns. However, the station data used in CHIRPS is mainly a monthly total from a limited number of stations. For the sake of independence it would have been advisable to exclude those stations from our validation data set. However, observed station data were not included consistently through the study period and even in a single year (e.g., stations used in the first month are not all included in second month). For example, in Ethiopia, in Jan/1983 monthly data from 140 stations are included in CHIRPS and decreased to 133 in Feb/1983. In addition, in Aug/2005 data from 213 stations are included in CHIRPS and decreased to 169 in Dec/2005. In addition to the publicly available data from Ethiopia, more data from hardcopies had been added to the available stations during a working visit at the National Meteorological Agency of Ethiopia. Therefore, for our validation procedure we used quality-controlled and improved/extended daily data from as many stations as possible.  For this validation data set we argued that the "dependency (is) rather weak and indirect" due to the much higher number of stations and the higher temporal resolution.

- I do think that CHIRPS is a very good product (from prior monthly/season scale evaluations and performance in hydrologic models & compared to other remotely sensed data), and it does need to be more carefully evaluated at the daily time-step.

Authors´ response:

- Thanks for pointing out the need for more careful evaluation at a daily time step. This is a very important point and that is why we tried to use the maximum possible validation areas and to find other methods than the typical point-pixel comparison. Using non-aggregated daily data resulted in a comparably weak correlation, but our 3$^{rd}$ method, stations average to area grid-cell average produces a good correlation. Therefore, our recommendation for hydrological (or other impact-) modelling is to use the area average (grid-cell average) instead of point/pixel information. We are using the same approach to model a water balance in one of the biggest rivers basins

in Ethiopia (area > 62,100 KM$^2$) and we found a good preliminary model performance of an r2=0.74, NSE=0.73 (final results will submitted to a journal).

- The station data that goes into the different rainfall products needs to be described in the methods/data.

Authors´ response:

- In Ethiopia, monthly data from 140 (January 1983) to 169 (December 2005) stations are included in CHIRPS. Additionally, in Kenya and Tanzania monthly data from 142 (Kenya) and 171 (Tanzania) in January 1983 and 62 (Kenya) and 55 (Tanzania) in December 2005 are included, respectively. We will include this information in the revised version as far as possible in section 2.2 (data sets) under CHIRPS page 7 line 21.

- In addition to the discussion. CHIRPS includes stations from several sources including GTS and GSOD, ARC includes GTS. Please include information on what stations the other products blend in.

Authors´ response:

- Thanks, ORH also used quality-controlled and gap-filled Global Summary of the Day (GSOD). This is already included in the paper (page 8, line 2), but to highlight the use of gap filled GSOD data we will modify the text as: *ORH is corrected for temporal inhomogeneity and biases and random errors are omitted through assimilation with quality checked GSOD data (Chaney et al., 2014).*

- The authors indicate that GSOD is only used in the CHIRPS monthly totals making the "dependency rather weak and indirect" Seems to me incorporating GSOD would contribute to the strong monthly correlations in Figure 4.

Authors´ response:

- As described in the paper, *"The inclusion of monthly station data can be assumed to improve CHIRPS´ performance compared to other rainfall products"*, so indeed we agree that inclusion of observed monthly totals will contribute to strong correlation on monthly time scale. While the correlated data are not fully independent, even at monthly resolution they are only partially related given that we re-calculated monthly means after quality-controlling daily values and used a much higher number of stations with more data added from hardcopies.

- From my interpretation of Funk et al. (2015) the GSOD data is included for pentad-totals as well. You may want to ask the data producers to clarify (and then include that information in the data/methods here.

Authors´ response:

- It is true that sparse GSOD data is also included in pentad-totals globally and we will add this information in section 2.2 (data sets) page 7 lines 9-21 as recommended.

- I* think* Ethiopia NMA stations are included in CHIRPS. Check with the date providers Funk et al. 2015 says: "Addi-tional observations have been provided by national meteorological agencies, primarily in Mexico, Central America, South America, and sub-Saharan Africa".

Authors´ response:

- Yes multiple stations, particularly monthly data, from Ethiopia are included in CHIRPS. But, as explained above, not all stations used in this study are included in

CHIRPS and the stations are not consistently used in the development of CHIRPS due to missing values. For example, in Jan/1983 monthly data from 140 stations are used and decreased to 133 in Feb/1983 and in Aug/2005 213 stations are used and decreased to 169 in Dec/2005. We will add more information about this in the revised version as described above.

- apparently ORH also uses GSOD "assimilating quality-controlled and gap-filled Global Summary of the Day (GSOD) in situ measurement"

Authors´ response:
- Yes GSOD data is used in ORH and this is included in the paper (page 8, line 2) to be clearer we will modify the text as described above.

- ...what spatial interpolation method do they use?

Authors´ response:
- A bilinear interpolation method is used and this will be added in (page 7, line 24) as: *ORH is developed by a spatial downscaling of the NCEP–NCAR reanalysis to a spatial resolution of 0.1º using a bilinear interpolation…..*

- Evaluation of daily rainfall for trend/variability/extremes/hydro model input is a worthwhile goal. Also not sure if the authors accomplished this given that i have questions about their metrics. I understand that you are comparing to stations but... Daily rainfall intensity: intensity is depth per unit time. How are you getting this when you just have daily totals? And then how does the "intensity" metric differ from what you describe as daily totals? Please include your definition of intensity.

Authors´ response:
- The analysis of trends/variability/extremes will be subject of another paper. We used the term average daily intensity to indicate average daily totals (mm/day) and we will change this into average daily rainfall (mm/day) in the revised version.

- Number of wet/dry days: is this just a count that does/does not match the stations? Or are you using something like probability of detection and false alarm rate? These metrics need to be defined in the methods.

Authors´ response:
- It is true that the number of wet/dry days is a count of daily records and we will add the definition as recommended in the revised version in section 3.2, page 11.

- I can't really tell what you did to come up with the results on page 13. Not obvious what "point to area-grid-cell" average means. I gather that its the average over the polygons shown in Figure 1, but this needs more explanation in the methods.

Authors´ response:
- Yes it is true "area-grid-cell average" means the basin/polygon average we will explain this in detail as recommended in section 3.2 page 9 line 21.

- Where do these polygons come from? Is there a reason why this level of basin was used to define the watersheds for a country?

Authors´ response:
- The polygons are basins retrieved from the global river basins available at the WaterBase hosted by the United Nations University (UNU-INWEH:

*http://www.waterbase.org*) and we will add few lines about the polygons and the data source in the revised version in section 3.1, page 8 lines 23-24.

- Since you're not comparing to hydrological/ streamflow data why not just average from 0.05 to 0.25 degree – essentially producing the same results as what you discuss with the coarser CHIRPS data?

Authors´ response:
- That is possible, but our objective is to find finer spatial resolutions that can be used later for hydrological and climate modelling in areas of the region with no ground observation. As we showed in section 4.2, page 15 lines 1-6, the improved version of CHIRPS (0.05 degree) is more accurate than the coarser resolution of CHIRPS (0.25 degree).

Specific comments (Minor)
- Additional information on how the data is produced should help explain your results (e.g. why is point to area-average best, does this have to do with the interpolation schemes that ARC and CHIRPS and the other product use?

Authors´ response:
- This is not because of the interpolation schemes that the products used but due to the method we used to compare the products. Compared to the point to pixel method, area averaging produces higher correlation and lower errors. During area averaging extremely high rainfall events obtained for a location from the various data products are levelled off by averaging and this makes the product much more accurate. In most of the rainfall products, including CHIRPS, there are occasionally higher daily rainfall values recorded and the averaging removes those extremes, which are much higher than the observed data in the area. We will include this information in the revised version in page 17 lines 1-3.

- Do CHIRPS results improve at 0.25deg because that is its original resolution, before being downcaled to 0.05deg with the CHPclim? This kind information will be useful for the other products as well.

Authors´ response:
- NO, there seems to be a misunderstanding. As we showed in section 4.2 page 15 lines 1-6, the higher resolution of CHIRPS (0.05 degree) showed an improvement in correlation with station data by up to 3.2% compared to the coarse resolution of CHIRPS (0.25 degree).

- Is only a historic record needed for env. Management? ORH isn't updated regularly (2012?) This should be clear in the paper, pls include in methods.

Authors´ response:
- We will include more detailed explanations and discussions on data availability and usage in the revised version. And yes, for environmental management real-time observations and projections of future climatic conditions are more important than historic records. We are working on this issue and added a respective section in the discussion.

- Meanwhile, ARC & CHIRPS are updated regularly. It will help contextualize the metrics if you discuss the products strengths and weaknesses more with some example of an environmental management application that they might be used for. I

am sure there are some that would benefit from ORH long record, or ARC's 1-day latency.

Authors´ response:

- Thank you very much for this remark and proposal. We will include the strength and weakness of the products in terms of resolution, length of time period and progress (regular updating) in the data set (section 2.2) and discussion part of the revised version.

- If you are including OHR why not include what they use routinely in the Africa Flood and Drought monitor? 3B42RT...

Authors´ response:

- We agree that the 3B42RT is a very good rainfall product and lots of papers are already available, but for our project we are interested in products with an observation period of more than 30 years, which is not the case for 3B42RT.

- how does blending datasets impact the application to environmental management? With respect to hydrologic modeling GLDAS (Rodell et al. 2004) uses ORH/Princeton+other, Africa flood and drought monitor (Sheffield et al. 2014) uses ORH/TRMM-RT, FLDAS (McNally et al. 2017) uses CHIRPS. How does all this relate to the climate models?

Authors´ response:

- This is a very good point and the products have shown a positive impact in environmental management in data sparse regions such as Africa. Compared to climate models, satellite based rainfall products, based on our findings, show higher accuracy (see figure 4) and we believe that satellite based products can be more accurate for environmental management than output from climate models. The role of data blending, including the recommended papers, in environmental management will be highlighted in the introduction part of the revised version.

- intro was vague and too focused on data scarcity - we have lots of data (models, remote sensing, some in situ)...just not lots of dense rainfall stations.

Authors´ response:

- Thank you very much; we will specify this in the revised version, pointing out that data scarcity mainly refers to station data.

- …there are lots of datasets to get temperature (e.g. MERRA-2, CFS-R). Why weren't these included?

Authors´ response:

- We screened multiple data sources for temperature and our selection is based on their spatial resolution, length of time period (> 30 year) and recommendations from previous papers.

- Technical corrections > fix citation (also 2017): Kimani, M., Hoedjes, J. and Su, Z.: Uncertainty Assessments of Satellite Derived Rainfall Products, , 15. doi:10.20944/preprints201611.0019.v1, 2016. Don't cite the pre-print use this one: Kimani, Margaret Wambui, Joost CB Hoedjes, and Zhongbo Su. "An Assessment of Satellite-Derived Rainfall Products Relative to Ground Observations over East Africa." Remote Sensing 9.5 (2017): 430.

Authors´ response:

- Thank you very much and this will be fixed in the revised version.

- Typo RFE pg 6...its RFE Rainfall Estimation Version 2 (REF 2.0) (Novella et al., 2013)

Authors´ response:
- Thank you very much and this will be fixed in the revised version.

References:

Chaney, N. W., Sheffield, J., Villarini, G., Wood, E. F., Chaney, N. W., Sheffield, J., Villarini, G. and Wood, E. F.: 20 Development of a High-Resolution Gridded Daily Meteorological Dataset over Sub-Saharan Africa: Spatial Analysis of Trends in Climate Extremes, Httpdxdoiorg101175JCLI--13-004231 [online] Available from: http://journals.ametsoc.org/doi/abs/10.1175/JCLI-D-13-00423.1 (Accessed 30 November 2016), 2014.

Dembélé, M. and Zwart, S. J.: Evaluation and comparison of satellite-based rainfall products in Burkina Faso, West Africa, Int. J. Remote Sens., 37(17), 3995–4014, doi:10.1080/01431161.2016.1207258, 2016.

Duan, Z., Liu, J., Tuo, Y., Chiogna, G. and Disse, M.: Evaluation of eight high spatial resolution gridded precipitation products in Adige Basin (Italy) at multiple temporal and spatial scales, Sci. Total Environ., 573, 10 1536–1553, doi:10.1016/j.scitotenv.2016.08.213, 2016.

---

## Author Response (AR1)

Reply to Anonymous Referee #1

We thank the reviewer for reviewing our manuscript and providing his/her valuable feedbacks. We have now addressed all of his/her comments and discussed them in the following. The comments were very helpful to identify some unclear issues with regard to the scope, methodologies and conclusions of the paper. We have revised the manuscript to resolve these issues and make our approach and conclusions more clear-cut. Thanks to the reviewer's feedback, the paper is now much improved.

General Comments
- This paper compared different rainfall datasets over East Africa that have 30+ year record, to station data in Ethiopia, Kenya and Tanzania. The authors found that the CHIRPS rainfall and the ORH Tmin/Tmax are the best products to use for long-term climate studies (trend, variability, and extreme indices) and input for climate or hydrological models. While I think this paper is a good start to necessary analysis of daily rainfall products, I have concerns with the lack of independent station data and the narrow scope of the research (results are only regionally relevant, not very generalizable). I think there are ways to work around this problem of data validation in sparse-regions but the authors would need to reframe the paper and consider how to address the greater challenge of evaluating the quality of satellite rainfall in data sparse regions. The authors also need to be more transparent/detailed about their methods (metrics and data sets).

Authors´ response:
- We appreciate acknowledgement that our paper provides a good start for analysing rainfall products. We would rather propose to speak of scarcity of station data, not a general lack of independent station data. We are convinced that this kind of validation of climate data products can only be done at a regional level and indeed we feel that it is impossible to do this at a significantly larger scale than provided here (3 countries in East Africa). We rephrased respective sections (abstract, page 2, lines 7-8), introduction (page 5, lines 19-25) and summary and conclusion (page 21, lines 2-4 and page 22 lines 3- 4 and 10-13) to make this point clearer. In fact, earlier studies with a similar goal of validating climate data (e.g., CHIRPS) products did so based on much more limited data basis in terms of temporal resolution or spatial dimension (e.g., *Duan et al., (2016) and Dembélé and Zwart (2016)*). We also think that our general approach is well generalizable and can and should be used for validating these climate data products in other areas as well.
  We added some explanations in the methodology part (section 3.1, page 9, lines 25-26 and section 3.2, page 10, line 26 to page 11, lines 1-3 and page 13, lines 3-10) to be more transparent and clear.

Specific Comments (Major)
- The focus on daily rainfall is useful/novel as the authors state that this has not been done before. In general this is a regional study that has limited applicability to studies beyond Ethiopia, Tanzania and Kenya, and to the extent that this is generalizable in country in questionable judging from Figure. 1. Not to say that research can't be done in data sparse regions but it has to be framed appropriately, and think that the authors could improve in this respect.

Authors´ response:
- Thanks for pointing out the novelty of our study. It is true that the study is a regional study with a main focus in Ethiopia, Kenya and Tanzania. Following this we modified

sections of the paper (e.g., abstract, page 2, lines 7-8), introduction (page 5, lines 19-25) and summary and conclusion (page 21, lines 2-4 and page 22 lines 3- 4 and 10-13)) to emphasize the regional focus – while pointing out that the approach is applicable in other regions, too.

Concerning framing we not only revised sections explaining the regional focus, but also improved the emphasis on the temporal dimension (daily resolution) and the different modes of data validation, going beyond the typical point to pixel comparison (see below).

- In fact, "how to evaluate rainfall and temperature in a data sparse region?" is a good question, although i don't think comparing to a handful of stations that are not independent is necessarily the answer. Major concern is the use of the EMA and GSOD data for evaluation and the conclusion that CHIRPS is the best performing product.

Authors´ response:

- We felt the same and this is why we posed the question – in this case for East Africa. Concerning the number of stations we did our best to get data from more than just a handful of stations: At least for Ethiopia the data set we used represents the most comprehensive to date and the most comprehensive possible (based on a quality controlled dataset from the National Meteorological Agency of Ethiopia). For Kenya and Tanzania data availability is admittedly thinner, partly due to quite restrictive data sharing policies (as explained in the paper), but the available data in our view do allow making the case that we validated the considered data sources at a regional scale. Adding data from two more countries provides additional insights about the accuracy of these data products. The concern with regard to the independency of data is taken up below. Indeed, the inclusion of station data in CHIRPS may raise such concerns. However, the station data used in CHIRPS from Ethiopia, Kenya and Tanzania is mainly a monthly total from a limited number of stations. For the sake of independence it would have been advisable to exclude those stations from our validation data set. However, observed station data were not included consistently in CHIRPS through the study period and even in a single year (e.g., stations used in the first month are not all included in second month). For example, in Ethiopia, in Jan/1983 monthly data from 139 stations are included in CHIRPS and decreased to 132 in Feb/1983. In addition, in Aug/2005 data from 175 stations are included in CHIRPS and decreased to 169 in Dec/2005 (ftp://chg-ftpout.geog.ucsb.edu/pub/org/chg/products/CHIRPS-2.0/diagnostics/). In addition to the publicly available data from Ethiopia, more data from hardcopies had been added to the available stations during a working visit at the National Meteorological Agency of Ethiopia. Therefore, for our validation procedure we used quality-controlled and improved/extended daily data from as many stations as possible. For this validation data set we argued that the "dependency (is) rather weak and indirect" due to the much higher number of stations and the higher temporal resolution. To make this clearer we added a detailed explanation in section 2.2 (data sets) in page 8, lines 19-27 and page 9, lines 1-6.

- I do think that CHIRPS is a very good product (from prior monthly/season scale evaluations and performance in hydrologic models & compared to other remotely sensed data), and it does need to be more carefully evaluated at the daily time-step.

Authors´ response:

- Thanks for pointing out the need for more careful evaluation at a daily time step. This is a very important point and that is why we tried to use the maximum possible validation areas and to find other methods than the typical point-pixel comparison. Using non-aggregated daily data resulted in a comparably weak correlation, but our 3$^{rd}$ method, stations average to area grid-cell average (explained below) produces a good correlation. Therefore, our recommendation for hydrological (or other impact-) modelling is to use the area average (grid-cell average) instead of point/pixel information. We are using the same approach to model a water balance in one of the biggest rivers basins in Ethiopia (area > 62,100 KM$^2$) and we found a good preliminary model performance of an r2=0.74, NSE=0.73 (final results will submitted to a journal).

- The station data that goes into the different rainfall products needs to be described in the methods/data.

Authors´ response:
- Under each product in section 2.2 (data sets), we added more information about the station data included in each product. For example, in page 7, lines 16-20 and page 8, lines 18-27 and page 9 lines 1-6 for CHIRPS and in page 8 lines 6-10 and lines 12-18 for ORH.

- In addition to the discussion. CHIRPS includes stations from several sources including GTS and GSOD, ARC includes GTS. Please include information on what stations the other products blend in.

Authors´ response:
- Thanks, ORH also used quality-controlled and gap-filled Global Summary of the Day (GSOD). This was already included in the paper (page 8, lines 12-15), but to highlight the use of gap filled GSOD data we modified the text as: *ORH is corrected for temporal inhomogeneity and biases and random errors are omitted through assimilation with quality controlled and gap filled ground station data available at NCDC (https://www.ncdc.noaa.gov/) as a global summery of the day (Chaney et al., 2014).*

- The authors indicate that GSOD is only used in the CHIRPS monthly totals making the "dependency rather weak and indirect" Seems to me incorporating GSOD would contribute to the strong monthly correlations in Figure 4.

Authors´ response:
- As described in the paper, *"The inclusion of monthly station data can be assumed to improve CHIRPS' performance compared to other rainfall products"*, so indeed we agree that inclusion of observed monthly totals will contribute to strong correlation on monthly time scale. While the correlated data are not fully independent, even at monthly resolution they are only partially related given that we re-calculated monthly means after quality-controlling daily values and used a much higher number of stations with more data added from hardcopies as explained above and more explanation is given in section 2.2 (data sets) in page 8, lines 19-27 and page 9, lines 1-6.

- From my interpretation of Funk et al. (2015) the GSOD data is included for pentad-totals as well. You may want to ask the data producers to clarify (and then include that information in the data/methods here.

Authors´ response:

- It is true that sparse GSOD/GTS data is also included in pentad-totals globally to produce a preliminary rainfall with a latency of two days and we added this information in section 2.2 (data sets) page 7 lines 16-20 as recommended.

- I\* think\* Ethiopia NMA stations are included in CHIRPS. Check with the date providers Funk et al. 2015 says: "Addi-tional observations have been provided by national meteorological agencies, primarily in Mexico, Central America, South America, and sub-Saharan Africa".

Authors´ response:

- Yes multiple stations, particularly monthly data, from Ethiopia are included in CHIRPS. But, as explained above, not all stations used in this study are included in CHIRPS and the stations are not consistently used in the development of CHIRPS due to missing values. For example, in Ethiopia, in January 1983 monthly data from 139 stations are used and decreased to 132 Stations in February 1983; in August 2005 about 175 stations are used and decreased to 169 in December 2005. Additionally, after 2005 the numbers of stations used in CHIRPS are declining and finally go to below 10 in 2015. We added more information about this in the revised version as described above in section 2.2 page 8 (lines 19-27) and 9 (lines 1-6).

- apparently ORH also uses GSOD "assimilating quality-controlled and gap-filled Global Summary of the Day (GSOD) in situ measurement"

Authors´ response:

- Yes GSOD data is used in ORH and this is included in the paper (page 8, lines 12-15) to be clearer we modified the text as described above *(ORH is corrected for temporal inhomogeneity and biases and random errors are omitted through assimilation with quality controlled and gap filled ground station data available at NCDC (https://www.ncdc.noaa.gov/) as a global summery of the day (Chaney et al., 2014)).* In addition, we added more information about the type of data included in ORH in page 8 lines 6-10.

- ...what spatial interpolation method do they use?

Authors´ response:

- A bilinear interpolation method is used and this is added in (page 8, lines 4-6) as: ORH is developed by a spatial downscaling of the NCEP–NCAR reanalysis (Kalnay et al., 1996) up to a spatial resolution of 0.1° using a bilinear interpolation. .....

- Evaluation of daily rainfall for trend/variability/extremes/hydro model input is a worthwhile goal. Also not sure if the authors accomplished this given that i have questions about their metrics. I understand that you are comparing to stations but... Daily rainfall intensity: intensity is depth per unit time. How are you getting this when you just have daily totals? And then how does the "intensity" metric differ from what you describe as daily totals? Please include your definition of intensity.

Authors´ response:

- The analysis of trends/variability/extremes will be subject of another paper. We used the term average daily intensity to indicate average daily totals (mm/day) and we changed this into average daily rainfall (mm/day) throughout the manuscript in the revised version.

- Number of wet/dry days: is this just a count that does/does not match the stations? Or are you using something like probability of detection and false alarm rate? These metrics need to be defined in the methods.

Authors´ response:
- It is true that the number of wet/dry days is a count of daily records and we added the definition as recommended in the revised version in section 3.2, page 13 lines 3-9. For your information, we also computed the probability of detection (POD) and false alarm rate (FAR), but it is not included here as it is not common for climate models. Comparing the satellite based rainfall products CHIRPS and ARC2 showed a higher agreement. For example for EthioShed1 we found a POD of 0.83, 0.84, and 0.49 and FAR of 0.12, 0.1, and 0.24 for ARC2, CHIRPS and ORH, respectively.

- I can't really tell what you did to come up with the results on page 13. Not obvious what "point to area-grid-cell" average means. I gather that its the average over the polygons shown in Figure 1, but this needs more explanation in the methods.

Authors´ response:
- Yes it is true "area-grid-cell average" means the basin/polygon average and we added an explanation in section 3.2 (methodology) page 10 (line 26) – page 11 (lines 1-3) as recommended.

- Where do these polygons come from? Is there a reason why this level of basin was used to define the watersheds for a country?

Authors´ response:
- The polygons are basins retrieved from the global river basins available at the WaterBase hosted by the United Nations University (UNU-INWEH: http://www.waterbase.org) and we add few lines about the polygons and the data source in the revised version in section 3.1, page 9 (lines 25-26) – page 10 (line 1).

- Since you're not comparing to hydrological/ streamflow data why not just average from 0.05 to 0.25 degree – essentially producing the same results as what you discuss with the coarser CHIRPS data?

Authors´ response:
- That is possible, but our objective is to find finer spatial resolutions that can be used later for hydrological and climate modelling in areas of the region with no ground observation. As we showed in section 4.2, page 16 (lines 25-26) – page 17 (lines 1-2), the improved version of CHIRPS (0.05 degree) is more accurate than the coarser resolution of CHIRPS (0.25 degree).

Specific comments (Minor)
- Additional information on how the data is produced should help explain your results (e.g. why is point to area-average best, does this have to do with the interpolation schemes that ARC and CHIRPS and the other product use?

Authors´ response:
- This is not because of the interpolation schemes that the products used but due to the method we used to compare the products. Compared to the point to pixel method, area averaging produces higher correlation and lower errors. During area averaging extremely high rainfall events obtained for a location from the various data products are levelled off by averaging and this makes the product much more accurate. In most of the rainfall products, including CHIRPS, there are occasionally higher daily rainfall

values recorded and the averaging removes those extremes, which are much higher than the observed data in the area. We included this information in the revised version in page 15 lines 3-7.

- Do CHIRPS results improve at 0.25deg because that is its original resolution, before being downcaled to 0.05deg with the CHPclim? This kind information will be useful for the other products as well.

Authors´ response:
- NO, there seems to be a misunderstanding. As we showed in section 4.2, page 16 (lines 25-26) – page 17 (lines 1-2), the higher resolution of CHIRPS (0.05 degree) showed an improvement in correlation with station data by up to 3.2% compared to the coarse resolution of CHIRPS (0.25 degree).

- Is only a historic record needed for env. Management? ORH isn't updated regularly (2012?) This should be clear in the paper, pls include in methods.

Authors´ response:
- Yes we agree not only historical records are required. It is true that ORH is not updated regularly (last update in February 2016) and the global data is available from 1901-2012 at different resolutions. To make this clearer we added few lines and its application, with a list of publications, in environmental management in page 8, lines 3-18.

- Meanwhile, ARC & CHIRPS are updated regularly. It will help contextualize the metrics if you discuss the products strengths and weaknesses more with some example of an environmental management application that they might be used for. I am sure there are some that would benefit from ORH long record, or ARC's 1-day latency.

Authors´ response:
- Thank you very much for this remark and proposal. We included the strength and weakness of the products in terms of resolution, length of time period and progress (regular updating) in the data set (section 2.2). For ARC2, for example, we modified the text in page 7 lines 9-11 as: *The data set is updated regularly (last update March 2018) and it is available at a spatial resolution of 0.1º covering the period of 1983–2018.*

- If you are including OHR why not include what they use routinely in the Africa Flood and Drought monitor? 3B42RT...

Authors´ response:
- We agree that the 3B42RT is a very good rainfall product and lots of papers are already available, but for our project we are interested in products with an observation period of more than 30 years, which is not the case for 3B42RT.

- how does blending datasets impact the application to environmental management? With respect to hydrologic modeling GLDAS (Rodell et al. 2004) uses ORH/Princeton+other, Africa flood and drought monitor (Sheffield et al. 2014) uses ORH/TRMM-RT, FLDAS (McNally et al. 2017) uses CHIRPS. How does all this relate to the climate models?

Authors´ response:

- This is a very good point and the products have shown a positive impact in environmental management in data sparse regions such as Africa. Compared to climate models, satellite based rainfall products, based on our findings, show higher accuracy (see figure 4) and we believe that satellite based products can be more accurate for environmental management than output from climate models. The role of data blending, including the recommended papers, in environmental management are highlighted for each data set (section 2.2) and a text is added to highlight the role of satellite-station data blending techniques in page 14, lines 14-16 by giving an example of CHIRP and CHIRPS. .

- intro was vague and too focused on data scarcity - we have lots of data (models, remote sensing, some in situ)...just not lots of dense rainfall stations.

Authors´ response:
- Thank you very much; we modified the introduction by removing some sections in the revised version, and we pointed out that data scarcity mainly refers to station data in pages 3 (lines 21-26) - 4 (lines 1-3).

- …there are lots of datasets to get temperature (e.g. MERRA-2, CFS-R). Why weren't these included?

Authors´ response:
- We screened multiple data sources for temperature and our selection is based on their spatial resolution, length of time period (> 30 year) and recommendations from previous papers.

- Technical corrections > fix citation (also 2017): Kimani, M., Hoedjes, J. and Su, Z.: Uncertainty Assessments of Satellite Derived Rainfall Products, , 15. doi:10.20944/preprints201611.0019.v1, 2016. Don't cite the pre-print use this one: Kimani, Margaret Wambui, Joost CB Hoedjes, and Zhongbo Su. "An Assessment of Satellite-Derived Rainfall Products Relative to Ground Observations over East Africa." Remote Sensing 9.5 (2017): 430.

Authors´ response:
- Thank you very much and this is fixed in the revised version.

- Typo RFE pg 6...its RFE Rainfall Estimation Version 2 (REF 2.0) (Novella et al., 2013)

Authors´ response:
- Thank you very much and this is fixed in the revised version.

Reply to Anonymous Referee #2

We thank the reviewer for reviewing our manuscript and providing his/her valuable feedbacks. We have now addressed all of his/her comments and discussed them in the following. The comments were very helpful to identify some unclear issues with regard to the scope, methodologies and conclusions of the paper. We have revised the manuscript to resolve these issues and make our approach and conclusions more clear-cut. Thanks to the reviewer's feedback, the paper is now much improved.

General comments
- This paper is specifically validating the quality of three climatic variables coming from different satellites data-streams and models using scientifically proven quality validation methodologies. The three include rainfall, maximum temperature and minimum temperature. Being a research that has been done for the first time that I know of, the paper unravels the different quality of each of these datasets and with evidence provide great knowledge of which is the best among the 6 datasets for each variable. If further validate the same dataset with observed rainfall and satellite from weather stations. Though not conclusive, through this research, one can relate that the CHIRPS dataset is better for rainfall analysis in specific areas which have complex topography with a case study in three East African countries. While the ORH dataset works best for the Tmax/ Tmin variable. The paper specifically highlight the methods used and why and how each one is best.
  East Africa being a complex region of climate analysis. The paper seems to have limited itself to specific sites which might not fully represent the entire region. Despite having fewer observed data the sample areas of interest might limit the imagination of the complexity of the region.

Authors´ response:
- Thanks for pointing out/acknowledging the difficulties in performing this kind of studies. Indeed the observed data is very limited in terms of spatial coverage and length of time period. In addition, getting daily data from the meteorological agencies is not easy, particularly from Kenya and Tanzania due to their data sharing policy. Therefore, for the purpose of validation we used almost all the available stations (210), particularly in Ethiopia, with an elevation, as given in Table 1, from 400-2510 and average elevation of the validation areas from 520-2830 meters. The included station data from Kenya and Tanzania range in altitude from 1097 to 2328 masl. Therefore, based on the stations at different elevations we conclude that our data set (which is the most comprehensive for East Africa to date) reasonably represents the study area even in those parts were no ground observations are available.

- CHIRPS products seemed to work well in some areas while at the same time came in second in other areas. The author should try and indicate by what percentage in all the analysis done was CHIRPS top and if the percentage is worth representing the region as the best dataset.

Authors´ response:
- We concluded CHIRPS to be the best product for rainfall based on the overall analysis (daily-monthly), the multiple statistics used (Table 3 and Figure 4 and 5) and the result of the analysis of rainfall characteristics (Table 4, see below). Table 4 is added in the revised version page 15 and the text in lines 21-24 is modified as: *On average, over the 21 validation areas, CHIRPS captures well the number of wet days (-0.17 % deviation), average duration of wet (-13.4 % deviation) and dry periods (-*

*17.6 % deviation), total rainfall (-4.5 % deviation), average amount of wet periods (-17 % deviation), and average daily rainfall (-7.7 % deviation) (Table 4).* The values in the bracket show the percentage deviation from the observed data. As we are using multiple statistical methods and multiple time scales and metrics, providing one general percentage for a product might be not appropriate.

Specific comments
- In page 9 of the document the author mentions that "The quality of selected stations was checked and extremely high rainfall records during dry seasons were excluded." Through this statement it is not clear what is considered as a dry season and the reason for exclusion of such rainfall dataset remains hanging. Also in consideration of the same, extreme event such as flash floods may be recorded in a single days' rainfall.

Authors´ response:
- During quality control few data were removed with extremely higher rainfall events such as >480 mm/day preceding and following dry days. We have done the quality control with the meteorology-experts in the field (colleagues from National Meteorological Agency, Ethiopia) and these data had been identified as error in inputting the data. This information is included in the revised manuscript in page 10, line 9-11.

- A few questions to be asked are; Could the x,y decimal places affect the location of a given station ending up reporting a value for a wrong location? For example a station reading of 36.123456, -1.123456 might fall at a different location compared to a reading of 36.123, -1.123. In this reference were the station locations validated?

Authors´ response:
- Good point: It is well conceivable that stations are falsely located in the next grid box of the product especially if you have a product with very high resolution such as CHIRPS and ARC2. For this reason we used all available station information and the extracted data is validated. This is can be a problem if you are comparing station to pixel. But if you are using the station average to area grid cell average the change from 36.123456, -1.123456 to 36.123, -1.123 might not be a problem if they are located inside the validation area as you are taking the average.

- From this paper it is also not clear what the following terms refer to; Wet days, duration of wet days and average amount of wet periods this might be confusing since they all are represented by one unit which is days. For example, when we talk about wet days we say 10 days. If we talk about duration of wet days do we still say 10 days? The same applies to the average amount of wet periods.

Authors´ response:
- Thank you very much for pointing out this issue. The unit for number of wet days (count of wet days in a year) is days/year, for average duration of wet periods (the number consecutive wet days) is days and for average amount of wet periods is mm. The full description of the rainfall characteristics with their units is included in the methodology part (section 3.2, page 13, lines 3-9) of the revised version and the results of the rainfall characteristics are provided in Table 4.

- From the paper it is very clear that the author highlights CHIRPS as the best rainfall product while ORH as the best temperature product. CHIRPS comes out better than

the rest based on the characteristics described by the author in page 14 but the author has not conclusively stated by how much is CHIRPS better than all this other products if you compare all the statistical analysis done. The Author has only highlighted that "In general, the observed rainfall characteristics are well captured by CHIRPS compared to CHIRP, ARC2, ORH, RCM, and RCMs."

Authors´ response:

- Thank you very much and we provide a table in page 15, lines 21-24 with a summary statistics of the rainfall characteristics, with a percentage deviation from the observation, as given below (Table 4) in the revised version and the text in page 16 lines 10-13 is modified as: *In general, the observed rainfall characteristics are well captured by CHIRPS, with a percentage difference from the observation of -0.17 % to -17.6 % for number of wet days and duration of dry periods, respectively, compared to CHIRP, ARC2, ORH, RCM, and RCMs (Table 4).*

- While at the same time pointed out areas that ARC2 has performed better that CHIRPS and CHIRP. Regarding the above, in some instances such as EthioShed4 the CHIRP and CHIRPS have equal R squared while in some areas ARC2 came on top. Through the analysis of all the Sheds analysed what percentage of CHIRPS compared to the rest of the datasets was better.

Authors´ response:

- Yes it is true that both ARC2 and CHIRP have shown higher R squared, considering the biases and errors, in 2 and CHIRPS in 17 of the 21 validation areas. However, in terms of capturing the daily rainfall characteristics (Table 4) ARC2 showed higher deviations compared to CHIRPS. In EthioShed4, CHIRP and CHIRPS have an equal R squared, but in terms of biases CHIRP showed higher biases (data points below and above the regression line) compared to CHIRPS (most of the data points lie in the regression line) as shown in figure 4. We added an explanation in page 14 lines 13-14 in the revised version.

- Still in line with that there are some areas where all the R squared were between 0.13 and 0.55, is it possible to elaborate on why such cases occur? Is it the methodology used to model the datasets that limits the correlation with the station data?

Authors´ response:

- The small R squared values were mainly computed for the regional climate models (RCMs). Compared to the satellite based rainfall products (which already include ground observed data), RCMs have a coarse spatial resolution (~ 50 km) and during the downscaling process of the global climate models they include less local information such as topographical features, which makes them weak in synthesising local daily rainfall, particularly in topographically very complex regions.

- Another question of concern is what explains the equal value for CHIRP and CHIRPS as portrayed in EthioShed4?

Authors´ response:

- In EthioShed4 it is true that R square value is the same, but if you see the distribution of the data above and below the regression line there is a difference, which is explained as a bias (over- or underestimation). To compare both products it is also good to see figure 5 (Taylor-diagram), which shows the correlation and standard deviation of each product on monthly time scale. Therefore, in addition to the bias shown in figure 4, figure 5 also shows a deviation between CHIRPS (with slightly

better correlation and standard deviation on monthly time scale similar to figure 4) and CHIRP.

- In the introduction the paper highlights CHIRPS as a dataset that has both station and satellite data in it. Might this explain the high correlation?

Authors´ response:

- As we explained in the methodology and discussion part, station data are included in CHIRPS, ARC2 and ORH. But compared to ARC2 and ORH, a larger number of stations are included in CHIRPS. Therefore, on a monthly time scale, the high correlation can be true due to the inclusion of monthly station data in the development of CHIPRS. In the revised version we include a section in page 8 lines 19-27 and page 9 lines 1-6 in the data sets section highlighting the stations included in CHIRPS (see also replies to reviewer #1).

- Are the same stations in CHIRPS used to validate the CHIRPS product?

Authors´ response:

- Yes – but due to different data processing and inconsistent use of station data in CHIRPS, the data included there are not fully congruent to the station data we used in the correlation. Multiple stations, particularly monthly data, from Ethiopia are included in CHIRPS as shown in table one and discussed in the discussion part. But, not all stations used in this study are included and the stations are not consistently used in the development of CHIRPS due to missing values. For example, in Ethiopia, in Jan/1983 monthly data from 139 stations are included and decreased to 132 in Feb/1983. In addition, in Aug/2005 monthly data from 175 stations are included and decreased to 169 in Dec/2005 (ftp://chg-ftpout.geog.ucsb.edu/pub/org/chg/products/CHIRPS-2.0/diagnostics/), which shows the inconsistency in the inclusion of the stations. We added more information in section 2.2 (data sets) in page 8 lines 19-27 and page 9 lines 1-6

In conclusion to the specific comments.

- The paper is very clear on how the validation is done. However, more can be done to ensure that these products are regarded as the best products as indicated by the author.

Authors´ response:

- Thank you very much for the comment and we consider your comments and we added a summary statistics table (e.g., Table 4) as mentioned above in the revised version.

- The paper currently is validating the products for areas with low observed dataset. Perhaps, the author can use historical analysis as a means of validation too. Also, an elaborate point of validation would be to highlight how the non-blended datasets such as CHIRP is performing compared to observed station data in regions that have well established network of weather stations such as the developed countries. Then further, validating the CHIRP against the CHIRPS. This will basically ensure less redundancy.

Authors´ response:

- Thank you very much and we added an explanation about the role of the station blending technique in page 14 line 14-16 in the revised version as recommended. Concerning the use of historic data: we did that to the extent possible, limited by the lengths of available time series, but in general using 30-year datasets. However, it would go beyond the scope of this paper to extend the analysis to further (developed) countries and regions. Adding even more data, figures and respective discussions to

the paper does not seem feasible to us. We definitely agree that it would be worthwhile to use the same approach of validating climate data products in other countries and regions. It is well conceivable that obtained results in terms of best products might look different, which could be due to various factors: higher spatial resolution, better general data quality, higher homogeneity of the region in terms of topography etc. To conclude: we feel that extensions to further regions and ultimately to the global scale requires (a) separate study/studies.

Technical comments

- In page 7, the Dekadal should come after pentadal since the former represents 10 days and the later represents 5 days.

Authors´ response:

- Thank you very much and we fixed this in the revised version.

- In page 13, 17 is numerical while three and one are text – you might want to use either for all.

Authors´ response:

- It is very common to convert numbers <10 to text, but not larger numbers. This will be checked with editorial policies of HESS.

- In Page 20, it is indicated that "The products are available with higher spatial and temporal resolution and for longer periods." – doesn't longer periods mean the same as temporal resolution?

Authors´ response:

- No, the terms have different meanings. Temporal resolution is used to indicate the time scale such as daily, dekadal and monthly; longer periods refers to the length of the time period/series such 30 or >30 years.

[revised manuscript text omitted]

---

## Referee Report (RR1)

**Evaluation of Multiple Climate Data Sources for Managing Environmental Resources in East Africa**

General comments about this manuscript, all the changes that had been requested earlier have been effected or well explained in the document. Additional information has been provided to ensure that most initial gaps are covered and all evidence presented forth by myself as a referee have been greatly explained. One good example is on page 10 where one can now understand why there are some values that were excluded in defining consecutive dry days. (>480mm). Also on page 15, It is also evident that CHIRPS compared to the rest of the datasets, given the statistics, is top in terms of rainfall analysis. The same applies to ORH as the best temperature dataset.
The manuscript goes deeper to elaborate on why CHIRPS and ORH leads in terms of the best datasets for Climate analysis (Rainfall and temperature) through its Validation and discussion topics. The methodology and sampling methods have been well explained throughout the manuscript.

In conclusion, the scientist have done great research on these datasets and provided enough evidence as to why CHIRPS and ORH tops the list. However there is always room for improvement given the challenges highlighted in terms of tackling or dealing with topographical issue and the diversity that East Africa provides. This, however, does not limit myself from approving this manuscript the next step.

---

## Author Response (AR2)

Reply to Anonymous Referee #1

We thank the reviewer for his/her valuable feedbacks. We have addressed all of his/her comments and discussed them in the following. Thanks to the reviewer's feedback, the paper is now much improved. In all your comments please kindly consider our objectives (in particular, daily resolution as well as high spatial resolution) in selecting data products.

- Overall, the paper has improved, but I don't think that the paper is ready for publication. I have provided some additional comments here that i think would make the paper more publishable in a journal like HESS. Overall, given the stated goal of being a rainfall evaluation i did not find it to be very comprehensive - Why didn't they include additional rainfall datasets like TAMSAT? ENACTS? (No citation of Tufa Dinku are surprising in an africa rainfall evaluation paper).

Authors´ response:
- As stated in the introduction, our goal was to evaluate climate (not only rainfall) data products at the highest possible temporal (i.e. daily) and spatial resolution. Given these boundary conditions, we consider our study to be comprehensive to the extent possible.
- Thanks for the recommendation to include datasets such TAMSAT and ENACTS. We are aware of these and other products like TAMSAT African Rainfall Climatology And Time series (TARCAT), or the Tropical Rainfall Measuring Mission (TRMM). The reason for not including these products in this evaluation is mainly due to the following factors, which we now explain more explicitly in the manuscript (page 5, lines 20-24 and page 7, lines 4-15):
    1. Temporal resolution: we only considered products available at **daily time scale,** but ENACTS and TAMSAT African Rainfall Climatology And Time series (TARCAT) are only available from dekadal (10 days) time scales
    2. Missing values: TMSAT has lots of known missing values, summing up to about 19 months during our analysis period (1981-2005), explained below. Therefore, we decided not to include this product as we have hydrological models not accepting missing values
    3. Length of the time period: products should cover a time period of more than 30 years. Therefore, products such as the TRMM are not included even if they are available at daily time scale.
    4. Spatial resolution: Selecting products with high spatial resolution helps to detect changes in climate and allow hydrological modelling at smaller local scales (e.g., smaller watersheds).
- We checked papers of Tufa Dinku and added accordingly

- And why not include additional temperature datasets like MERRA-2, CFS-R? I recognize that one can't compare all products, but then it needs to be justified how these product are chosen e.g. relevant to a larger project.

Authors´ response:

- We agree that evaluating all the available products is not feasible at least in one paper including 3 variables. Products such as the Observational Reanalysis Hybrid (ORH), MERRA-2, and CFS-R are all based on the reanalysis data. Compared to MERRA-2 (~ 50 km) and CFS-R (~ 38 km), ORH is available at higher spatial resolution (up to ~10km) and it includes more local climate information and it has been widely used in climate and hydrological studies including in East Africa as given on page 8 lines 15-28. Therefore, based on the spatial resolution, inclusions of local climate information, and previous studies, we decided to use only ORH in this analysis. To acknowledge the other products, we added a section on page 4 lines 21-24 and the selection criteria on page 5 lines 20-24 and page 7 lines 14-15.

- The paper mentions the need for daily data for hydrologic modeling, and CORDEX for climate applications but did not include either of these applications. So I would suggest either making the rainfall evaluation more comprehensive (more comments below) or potentially adding an application piece.

Authors´ response:
- We think there is a misunderstanding here. First, we included RCM (which includes CORDEX). Second, we are not saying CORDEX is only suitable for climate application. As we explained in the discussion part on page 21, lines 8-19, the performance of RCMs is weak for rainfall as they show higher biases and errors compared to the other rainfall products. Therefore, we conclude that if RCMs are used in local scale hydrological and climate models they should be bias corrected using the most commonly used bias correction techniques such as the empirical quantile mapping. When we say climate models, we mean regional or global climate models, but also the statistical models (e.g., SDSM) that are commonly used in local scale climate studies. These model require long-term daily climate data (e.g., precipitation and temperature) to produce local scale climate change information as explained on page 6 lines 3-6. To make this clearer we added SDSM as an example in the text on page 21 line 19.
- Concerning the application piece, we definitely agree that it should be done, actually we are currently working on it. However, we also feel that including it into this paper is not feasible, since it goes far beyond the scope of this paper.

- I do think that the daily analysis is novel, interesting and should be published, as well as their careful consideration of the point to pixel problem. But in the current presentation is not sufficient for HESS publication.

Authors´ response:
- We appreciate the acknowledgment that the paper provides novel and interesting analysis. We also thank for the recommendation to further improve our manuscript. Below, we provide a point by point reply to your comments.

Major Comments:

- Page 7 line 9: Example of a not very clearly written sentence. I would suggest having an outside reader (or co-author) review the paper for clarity. I have trouble following this section. For example: CHIRPS is a satellite-gauge blended product. The CHIRPS-prelim is updated every 5 days, at 2 day latency (not sure if that is 100% accurate) and blends in GTS stations. The CHIRPS-final product, updated around the 20th of each month blends in GTS, GSOD and other local or regional stations (including some NMA) that the product developers are able to acquire. For the final product, these stations are used to adjust monthly totals.

Authors´ response:
- We modified the section according to Funk et al.,( 2015) to make it clearer on page 8 lines 2-7.
- Page 7, line 15: Last update Feb 18 (should say: to present), and 1981-2018 (should also say "to present") since Feb 2018 won't be accurate by the time this paper is published.

Authors´ response:
- Thank you for the recommendation and the text is now modified on page 7 lines 22-24 and page 8 lines 8-10.

- I appreciate that the authors tried to add the explanation page 8 lines 10-25, but it was not very clearly written and using the original data from NMA doesn't necessarily make it independent --since some of those stations should be in the CHIRPS.

Authors´ response:
- This part is now modified to make it clearer on page 9 lines 3-24. Basically, the reviewer is right in pointing out that using original (daily) data from NMA does not necessarily make it independent. However, we see no way out of this dilemma that the evaluated data is not fully independent, since the NMA data set is the only available comprehensive source for ground-truthing. As we tried to explain in the paper, NMA data blended into CHIRPS represent monthly values from only a limited number of stations (which is not consistent over time, see above), while the evaluation data set we used here is based on original daily data from a much higher number of stations. They were quality controlled as explained, that is, partly diverging from the data blended into CHIRPS. This means, overall the dependency should be weak and there is simply no other option to evaluate CHIRPS in this particular region.

- That said, the ENACT data said does contain many of the NMA stations. The authors should acknowledge that ENACTS dataset exists and include it in the analysis or justify why it was not included https://link.springer.com/content/pdf/10.1186%2F2194-6434-1-15.pdf

Authors´ response:

- Thanks for the recommendation. As a data source ENACTS, TAMSAT are mentioned in the introduction part (page 4 lines 14-19). Even though the spatial resolution and temporal coverage of both products fit our needs, they do not fulfil other essential criteria:
  - ENACTS is available only at dekadal (10 days) time scale and we only consider products available at daily time scale and

- o TAMSAT has considerable missing values (
  (Missing months: 1983-01 & -11; 1984-11; 1985-02, -03, & -11;1986-08; 1988-11 & -12; 1989-01, -02, -05 & -09; 1990-01 & -02; 1992-02; 1993-05; 1996-10; 1999-01; 2006-09; source https://icdc.cen.uni-hamburg.de/1/daten/atmosphere/tamsat-rainfall-africa/ and Maidment et al., 2017) which limits its applicability as many hydrological models don't accept missing values (e.g., BROOK90). This information is added in the revised version on page 5 lines 20-24 and page 7 lines 10-15.

- The authors include CORDEX but mostly say that it doesn't compare well with stations. But there are important applications for this dataset. So, the message in the paper that CORDEX rainfall isn't good isn't very useful since people look to CORDEX for its climate applications. What do the authors suggest for improvements?

Authors´ response:

- It is true that the CORDEX regional climate models (RCMs) are widely used for regional climate assessment (variability and trends on a large scale) as explained in the introduction part (page 5 lines 5-7), but they perform weakly in representing station based or local climate information particularly for daily precipitation. In climate modelling, precipitation is one of the most difficult variables to model, particularly at smaller scales and this makes it more difficult in topographically complex regions such as East Africa. Therefore, as we explained in in the discussion part (page 21 lines 8-19), precipitation data from RCMs should be improved by applying bias correction techniques before application in hydrological and climate models. As a conclusion we added a line as: "*In general, in topographically complex regions such as East Africa, RCMs require further improvements in terms of spatial resolution and accuracy by adding more local information in the modelling process, particularly for precipitation*". on page 21 lines 19-21.

Minor:

- Page 7 line 9: Stations data should be "station data"

Authors´ response:
- Thanks, this is modified (page 8, line 2)

- Line 10 page 15: 17, three, and one [use numbers in a list]

Authors´ response:
- Thanks, this is modified (page 16, line 11).

[revised manuscript text omitted]

---

## Author Response (AR3)

Reply to comments by the editor on "Evaluation of Multiple Climate Data Sources for Managing Environmental Resources in East Africa"

Comments to the Author:

Dear Authors,

Thank you for revising the manuscript and providing response to the reviewer #1. I think the manuscript is improved however I think a few more changes (moderate in nature) are needed. Please see my comments below. They mainly intend to improve upon the presentation of the results.

Authors´ response:

Dear Dr. Shukla,

Thank you very much for the valuable feedback and your support to improve the manuscript. We have addressed all of your comments and discussed them in the following.

- Please consider converting table 3, 5 and 6 into graphs. There are just too many numbers to keep track of there. It would be much easier to keep track of the relative performance of each of the datasets if these results were presented in a graphical format.

Authors´ response:

- Thanks for the recommendation. Tables (3, 5, and 6) are now converted into graphs (bar-plots) as follows for rainfall, T-max, and T-min.

[Figure]

*Fig. 3. Statistical evaluation of daily rainfall retrieved from ARC2, CHIRP, CHIRPS, ORH, and RCMs against ground observations: CC (top panel), RMSE (middle panel), and Rbias (lower panel) during the period of 1983–2005 for 21 validation areas of East Africa (see Table 1).*

[Figure]

*Fig. 4. Statistical evaluation of daily T-max retrieved from GFDL, HadGEM2, MPI, ORH, and RCMs against ground observations: CC (top panel), RMSE (middle panel), and Rbias (lower panel) over the period of 1983–2005 for 21 validation areas of East Africa (see Table 1).*

[Figure]

*Fig. 5. Statistical evaluation of daily T-min from retrieved GFDL, HadGEM2, MPI, ORH, and RCMs against ground observations: CC (top panel), RMSE (middle panel), and Rbias (lower panel) over the period of 1983–2005 for 21 validation areas of East Africa (see Table 1).*

- (2) Please use a different color bar for Figure 2. Please limit the range from 0 to 12 (why is there below 0 in a precipitation color scale?). Perhaps choose a color scale that is sequential going from a light color to a darker color. Also in this figure and in fig 3, it may be best to show absolute value of rainfall or temperature using one of the reference dataset and show only difference between other datasets and that reference dataset.

Authors´ response:

- Thanks for the recommendation. The figures, colours and colour bars, are changed as recommended. To show the absolute values of rainfall and temperature datasets, CHIRPS and ORH, respectively, are used as a reference dataset and section 4.1 is modified by presenting few results to select CHIRPS and ORH as a reference dataset. While selecting the colours we consider for colour-blind readers.

[Figure]

Fig. 6: Average daily rainfall (mm day-1) map of East Africa retrieved from CHIRPS (left panel) for the study period 1983–2005. For rainfall data from ARC2, CHIRP, ORH, and RCM (right panels) the absolute difference (mm day-1) from CHIRPS is displayed. All the maps are given in a 0.05° spatial resolution.

[Figure]

Fig. 7: Map of average daily T-max and T-min (°C) for East Africa generated from ORH (left panels) for the study period 1983–2005. For temperature data from HadGEM2, GFDL, and MPI (right panels) the absolute difference (°C) from ORH is displayed. All the maps are given in a 0.1° spatial resolution.

- (3) Table 4: I like this analysis but it would be useful to see how each of these the spatial variability of daily rainfall characteristics as indicated by different datasets. The datasets probably don't compare as well over the high altitude and rainfall regions than low altitude and low rainfall regions. It would be good see the spatial variability in those characteristics and how each of the dataset performs relative to observations.

Authors´ response:

- As shown in Table 4, most of the rainfall characteristics are well produced from CHIRPS. Looking into the individual validation areas, CHIRPS comes first followed by ARC2. For few rainfall characteristics ORH and RCMs showed a lower deviation but in some areas show very high deviations. Hence, taking the individual areas CHIRPS and ARC2 are the products with an overall lower deviation from the observed values.
- We added a table (given below) in the supplementary material (ST. 2) and a line explaining this issue is added on page 17 lines 1-2 as: "*Moreover, the observed daily rainfall characteristics are well represented by CHIRPS and ARC2 in most of the validation areas (ST. 2).*"

*ST. 2: The performance of individual rainfall products in representing the observed daily rainfall characteristics in the 21 validation areas. The values for CHIRP, ARC2, CHIRPS, Had, MPI, GFDL, and RCMs is given as a percentage of deviation from the observed values. Lowest deviations are marked in bold. Rainfall characteristics (RC) 1, 2,3,4,5 and 6 represent (1) Number of wet days (days/year), (2) Average duration of wet periods (days), (3) Total amount of precipitation (mm/year), (4) Average amount of wet periods (mm), (5) Average duration of dry periods (days), and (6) Average daily precipitation (mm/day), respectively.*

| Validation area | RC | Observed | CHIRP | ARC2 | CHIRPS | ORH | Had | MPI | GFDL | RCMs |
|---|---|---|---|---|---|---|---|---|---|---|
| EthioShde1 | 1 | 268.3 | -24.6 | 14.9 | 12.9 | 26.5 | 16.2 | 6.7 | 32.2 | **-4.7** |
| | 2 | 10.2 | -91.4 | 26.3 | 17.3 | 244.5 | -46.1 | -36.8 | **-14.9** | -44.8 |
| | 3 | 1824.4 | **9.2** | 43.0 | 15.6 | 24.6 | 76.2 | 57.6 | 61.9 | 64.8 |
| | 4 | 69.2 | -87.6 | 57.2 | 20.1 | 239.2 | -18.3 | -6.6 | **4.2** | -4.6 |
| | 5 | 3.7 | 154.9 | **-9.0** | -21.0 | 72.0 | -65.1 | -49.5 | -61.5 | -32.8 |
| | 6 | 6.8 | 44.8 | 24.5 | 2.4 | **-1.5** | 51.6 | 47.7 | 22.4 | 72.9 |
| EthioShde2 | 1 | 242.8 | -32.1 | 28.2 | 23.5 | 20.2 | 10.2 | **1.8** | 11.8 | -20.1 |
| | 2 | 8.4 | -96.4 | 12.2 | 20.4 | 204.9 | -23.9 | -26.8 | **-3.7** | -46.8 |
| | 3 | 985.0 | **-3.9** | 40.7 | 8.1 | -13.5 | -24.2 | -24.4 | -36.5 | -28.9 |
| | 4 | 34.2 | -94.9 | 23.0 | **5.4** | 119.6 | -47.7 | -45.6 | -45.3 | -52.7 |
| | 5 | 4.2 | 183.2 | -33.6 | -29.3 | 90.1 | -39.4 | -30.6 | **-28.7** | 32.5 |
| | 6 | 4.1 | 41.5 | **9.7** | -12.5 | -28.0 | -31.3 | -25.7 | -43.2 | -11.1 |
| EthioShde3 | 1 | 215.2 | -40.1 | 14.9 | **-8.3** | 15.3 | -9.0 | -12.6 | 19.9 | -24.6 |
| | 2 | 9.4 | -98.6 | **8.1** | 18.8 | 241.1 | -35.6 | -35.3 | 21.0 | -44.6 |
| | 3 | 1299.8 | 12.7 | 41.3 | 16.8 | 10.5 | -7.4 | -15.8 | 57.7 | **3.3** |
| | 4 | 57.1 | -97.4 | 32.9 | 51.4 | 227.1 | -36.6 | -37.6 | 59.2 | **-24.1** |

| | | | | | | | | | |
|---|---|---|---|---|---|---|---|---|---|
| | 5 | 6.4 | 369.2 | -15.8 | 58.1 | 141.8 | -24.0 | **-9.1** | -20.5 | 34.5 |
| | 6 | 6.0 | 88.2 | 23.0 | 27.4 | -4.1 | **-1.6** | -3.7 | 31.5 | 37.0 |
| EthioShde4 | 1 | 166.9 | -53.0 | 34.0 | **-3.6** | 4.9 | -27.5 | -31.0 | -23.3 | -47.4 |
| | 2 | 5.1 | -95.3 | 41.9 | **30.5** | 139.1 | -55.5 | -53.5 | -35.3 | -72.1 |
| | 3 | 944.0 | -3.2 | 60.5 | **0.2** | -1.1 | -30.7 | -33.0 | -25.4 | -30.0 |
| | 4 | 29.0 | -90.3 | 69.9 | **35.7** | 125.5 | -57.4 | -54.8 | -37.0 | -62.9 |
| | 5 | 6.1 | 296.5 | -7.5 | 47.5 | 118.1 | **-6.5** | 7.7 | 13.0 | 117.9 |
| | 6 | 5.7 | 105.8 | 19.8 | 4.0 | -5.7 | -4.5 | -2.9 | **-2.7** | 33.1 |
| EthioShde5 | 1 | 275.4 | -22.7 | 32.4 | 20.7 | 48.3 | 26.7 | 12.6 | 40.6 | **-1.4** |
| | 2 | 8.4 | -94.7 | 60.9 | 52.5 | 230.6 | -38.7 | -40.2 | **-20.7** | -47.0 |
| | 3 | 1046.8 | -6.9 | 29.0 | **1.8** | -11.3 | 49.1 | 22.1 | 3.3 | 21.8 |
| | 4 | 31.8 | -93.6 | 56.7 | **28.6** | 97.8 | -28.9 | -35.2 | -41.7 | -34.5 |
| | 5 | 2.7 | 73.8 | -23.4 | **-10.1** | 11.4 | -69.5 | -60.4 | -70.0 | -43.8 |
| | 6 | 3.8 | 20.5 | **-2.6** | -15.7 | -40.2 | 17.6 | 8.5 | -26.5 | 23.6 |
| EthioShde6 | 1 | 244.0 | -31.8 | 19.5 | **0.0** | 20.1 | 0.0 | -9.6 | -4.9 | -24.8 |
| | 2 | 7.3 | -96.9 | 43.1 | **17.4** | 156.8 | -48.2 | -51.4 | -51.2 | -68.2 |
| | 3 | 914.8 | -4.0 | 80.6 | **-1.8** | -14.4 | -40.2 | -45.5 | -68.6 | -55.3 |
| | 4 | 27.5 | -95.6 | 116.2 | **15.3** | 83.1 | -69.0 | -70.7 | -83.9 | -81.1 |
| | 5 | 3.6 | 131.8 | **-0.9** | 29.0 | 59.9 | -45.8 | -31.8 | -42.5 | 24.6 |
| | 6 | 3.7 | 40.7 | 51.1 | **-1.8** | -28.7 | -40.2 | -39.7 | -67.0 | -40.5 |
| EthioShde7 | 1 | 267.5 | -24.8 | 14.6 | **7.5** | 56.2 | 11.4 | 8.4 | 22.4 | -11.1 |
| | 2 | 9.5 | -92.4 | 43.2 | 32.3 | 344.8 | -42.8 | -32.1 | **-17.5** | -50.6 |
| | 3 | 1662.5 | **-3.3** | 42.1 | 4.8 | 15.8 | 20.2 | 7.6 | -10.5 | 4.1 |
| | 4 | 59.1 | -90.3 | 77.4 | **28.9** | 229.6 | -38.3 | -29 | -39.7 | -42.1 |
| | 5 | 3.5 | 130.1 | **4.1** | 13.9 | 43.2 | -58.1 | -37.6 | -55.0 | -15.2 |
| | 6 | 6.2 | 28.5 | 23.9 | **-2.5** | -25.9 | 7.9 | 5.1 | -26.8 | 17.2 |
| EthioShde8 | 1 | 289.3 | -19.2 | 16.3 | **4.9** | 45.1 | 10.7 | 5.2 | 24.6 | -9.2 |
| | 2 | 10.6 | -95.4 | 53.3 | 18.9 | 330.6 | -50.2 | -44.9 | **-15.8** | -59.6 |
| | 3 | 1778.8 | 3.5 | 26.7 | **2.4** | 23.5 | 2.5 | -18.8 | -37.6 | -21.3 |
| | 4 | 65.0 | -94.1 | 67.0 | **16.1** | 266.3 | -53.9 | -55.3 | -57.8 | -65.0 |
| | 5 | 2.8 | 110.9 | **-1.9** | 9.7 | 35.5 | -65.3 | -45.3 | -61.4 | -27.5 |
| | 6 | 6.1 | 28.0 | 8.9 | **-2.3** | -14.9 | -7.4 | -18.9 | -49.9 | -13.4 |
| EthioShde9 | 1 | 244.3 | -31.0 | 80.7 | 51.4 | 92.5 | 54.8 | 42.7 | 63.6 | **-45.0** |
| | 2 | 5.8 | -93.2 | 64.3 | 67.0 | 239.9 | **6.4** | 16.8 | 36.4 | -34.8 |
| | 3 | 813.1 | 36.6 | 33.2 | **30.0** | 50.4 | 182.5 | 100.8 | 45.3 | 93.2 |
| | 4 | 19.3 | -86.5 | **18.4** | 46.7 | 165.6 | 94.3 | 64.4 | 21.2 | 32.5 |
| | 5 | 2.9 | 98.7 | -48.9 | -30.8 | **-10.4** | -58.8 | -49.0 | -53.2 | -23.5 |
| | 6 | 3.3 | 97.9 | -28.0 | -12.1 | -21.9 | 82.5 | 40.7 | **-11.2** | 103.3 |
| EthioShde10 | 1 | 199.0 | -44.4 | 63.5 | **-0.3** | 20.3 | -10.8 | -15.9 | -7.5 | -34.7 |
| | 2 | 6.8 | -97.8 | 73.9 | 34.7 | 214.5 | -38.5 | -41.0 | **-26.1** | -61.1 |
| | 3 | 992.7 | **3.3** | 66.5 | 4.9 | -12.9 | -25.7 | -27.1 | -31.1 | -28.3 |
| | 4 | 33.9 | -95.9 | 77.1 | **41.8** | 127.7 | -48.8 | -48.9 | -44.9 | -57.2 |
| | 5 | 5.7 | 242.5 | -22.7 | 45.2 | 117.3 | -16.1 | **-9.4** | -11.2 | 64.0 |
| | 6 | 5.0 | 85.8 | **1.8** | 5.2 | -27.6 | -16.7 | -13.4 | -25.5 | 9.9 |

| | | | | | | | | | |
|---|---|---|---|---|---|---|---|---|---|
| EthioShde1 1 | 1 | 241.6 | -32.2 | 42.6 | 5.6 | **0.5** | 12.2 | -5.5 | 24.5 | -19.6 |
| | 2 | 6.0 | -96.1 | 59.7 | **15.6** | 76.4 | -36.7 | -49.8 | -22.2 | -68.2 |
| | 3 | 1001.4 | **2.4** | 68.6 | 3.3 | 12.2 | 257.3 | 83.1 | 63.1 | 107.7 |
| | 4 | 25.1 | -94.1 | 88.7 | 13.1 | 96.9 | 101.8 | -2.8 | **1.9** | -17.9 |
| | 5 | 3.1 | 114.3 | -23.6 | **7.9** | 73.6 | -51.7 | -40.1 | -54.8 | -24.7 |
| | 6 | 4.1 | 51.1 | 18.2 | **-2.2** | 11.6 | 218.5 | 93.8 | 31.0 | 158.3 |
| EthioShde1 2 | 1 | 32.0 | -90.3 | **-18.8** | -59.5 | -69.1 | -61.7 | -72.0 | -62.0 | -82.2 |
| | 2 | 1.8 | -90.2 | -27.1 | -15.3 | **7.0** | -40.7 | -50.5 | -43.4 | -68.3 |
| | 3 | 246.3 | -18.9 | 47.9 | -21.8 | -26.1 | 103.3 | 20.3 | -18.7 | **17.1** |
| | 4 | 14.2 | **-17.7** | 32.7 | 63.7 | 156.2 | 214.3 | 112.9 | 21.0 | 108.7 |
| | 5 | 18.6 | 1117.4 | **-3.9** | 153.1 | 342.9 | 86.6 | 136.2 | 76.9 | 225.3 |
| | 6 | 7.7 | 738.1 | **82.1** | 93.2 | 139.5 | 430.4 | 330.2 | 113.8 | 557.7 |
| EthioShde1 3 | 1 | 112.0 | -61.9 | **10.4** | -35.5 | -22.8 | -50.3 | -51.7 | -41.2 | -64.0 |
| | 2 | 3.7 | -98.0 | 24.8 | **0.7** | 88.4 | -64.2 | -64.3 | -37.5 | -77.3 |
| | 3 | 661.3 | 23.5 | 48.5 | 10.2 | **5.0** | -19.0 | -31.4 | -13.9 | -22.3 |
| | 4 | 22.0 | -93.4 | 67.9 | 72.1 | 156.4 | -41.7 | -49.3 | **-8.6** | -51.1 |
| | 5 | 6.6 | 323.9 | **-9.4** | 72.6 | 121.1 | 9.8 | 10.6 | 21.4 | 130.2 |
| | 6 | 5.9 | 224.0 | **34.6** | 70.9 | 36.1 | 62.8 | 42.2 | 46.3 | 115.6 |
| EthioShde1 4 | 1 | 145.5 | -59.7 | **0.6** | -24.9 | -10.2 | -21.4 | -29.4 | -15.3 | -48.3 |
| | 2 | 3.2 | -97.9 | **-14.5** | -24.9 | 50.0 | -54.7 | -57.7 | -37.6 | -72.2 |
| | 3 | 709.4 | 2.3 | 21.6 | **0.4** | -12.0 | 114.5 | 73.1 | 106.1 | 95.0 |
| | 4 | 15.8 | -94.7 | 3.3 | **0.4** | 47.0 | 23.5 | 3.7 | 51.9 | 4.7 |
| | 5 | 4.9 | 171.2 | **-9.3** | 36.9 | 80.6 | -27.6 | -17.4 | -16.3 | 40.2 |
| | 6 | 4.9 | 154.0 | 20.9 | 33.7 | **-2.0** | 172.8 | 145.1 | 143.3 | 277.1 |
| EthioShde1 5 | 1 | 80.9 | -76.9 | **-34.2** | -53.6 | -62.2 | -62.5 | -66.9 | -60.8 | -71.7 |
| | 2 | 3.1 | -95.3 | -14.0 | -15.9 | **3.4** | -76.1 | -77.5 | -68.9 | -79.3 |
| | 3 | 501.4 | **5.3** | 42.4 | 6.1 | -55.8 | -26.3 | -36.5 | -43.2 | -36.2 |
| | 4 | 19.2 | -78.5 | 86.1 | 92.5 | **20.7** | -53.0 | -56.9 | -54.9 | -53.4 |
| | 5 | 9.1 | 438.4 | 36.8 | 140.3 | 330.1 | **5.5** | 33.5 | 19.4 | 119.3 |
| | 6 | 6.2 | 356.8 | 116.3 | 128.9 | **16.8** | 96.5 | 91.5 | 44.8 | 125.5 |
| EthioShde1 6 | 1 | 164.3 | -53.9 | 14.6 | -21.9 | **-11.6** | -19.8 | -29.5 | -23.4 | -44.7 |
| | 2 | 3.7 | -97.3 | **-0.1** | -19.6 | 40.3 | -59.5 | -67.5 | -54.9 | -73.8 |
| | 3 | 822.5 | -8.2 | 68.6 | -12.2 | -17.0 | **4.2** | -8.7 | -31.9 | -15.1 |
| | 4 | 18.5 | -94.7 | 47.0 | **-9.6** | 31.8 | -47.4 | -58.0 | -59.9 | -59.7 |
| | 5 | 4.1 | 182.6 | **-22.8** | 31.2 | 62.3 | -40.2 | -36.2 | -28.2 | 27.2 |
| | 6 | 5.0 | 98.9 | 47.1 | 12.4 | **-6.0** | 29.8 | 29.4 | -11.0 | 53.5 |
| EthioShde1 7 | 1 | 251.6 | -29.8 | 20.8 | **1.9** | 17.6 | 16.6 | 3.1 | 21.9 | -12.0 |
| | 2 | 8.5 | -97.3 | 50.4 | 24.0 | 183.8 | -34.3 | -38.2 | **-14.6** | -42.4 |

| | | | | | | | | | |
|---|---|---|---|---|---|---|---|---|---|
| | 3 | 917.6 | -7.6 | 38.5 | **-3.4** | -19.2 | 34.9 | 16.2 | 3.9 | 16.8 |
| | 4 | 30.9 | -96.5 | 72.5 | **17.6** | 95.0 | -24.0 | -30.3 | -27.2 | -23.6 |
| | 5 | 3.8 | 151.8 | **-0.6** | 28.8 | 81.1 | -55.6 | -43.8 | -49.7 | -6.5 |
| | 6 | 3.6 | 31.5 | 14.7 | **-5.2** | -31.3 | 15.6 | 12.7 | -14.8 | 32.7 |
| KenShed1 | 1 | 146.5 | -58.6 | -25.7 | **16.2** | -47.0 | -33.3 | -39.8 | -37.6 | -53.7 |
| | 2 | 3.2 | -96.5 | -19.1 | **-1.7** | -48.6 | -51.1 | -57.5 | -62.4 | -80.1 |
| | 3 | 554.6 | -30.7 | 21.7 | -14.3 | -42.6 | 58.1 | **0.8** | -13.7 | 6.5 |
| | 4 | 12.3 | -94.2 | 32.5 | -27.5 | -44.4 | **16.0** | -28.9 | -48.0 | -54.3 |
| | 5 | 4.8 | 215.7 | 56.1 | -22.8 | 143.9 | 15.5 | 27.9 | **2.2** | 103.1 |
| | 6 | 3.8 | 67.7 | 63.7 | -26.2 | **8.2** | 137.1 | 67.5 | 38.3 | 130.0 |
| KenShed2 | 1 | 180.5 | -48.1 | **-4.9** | 84.2 | -38.6 | -34.7 | -41.9 | -36.3 | -47.8 |
| | 2 | 3.4 | -92.9 | **-9.4** | 22.2 | -54.3 | -76.9 | -83.9 | -80.5 | -91.8 |
| | 3 | 1018.0 | -20.2 | 37.8 | 3.3 | -24.7 | **2.0** | -34.7 | -61.7 | -42.0 |
| | 4 | 18.9 | -89.1 | 31.2 | **-31.5** | -43.9 | -63.9 | -81.9 | -88.3 | -90.9 |
| | 5 | 3.2 | 91.5 | **3.3** | -56.7 | 83.2 | -26.1 | -11.6 | -34.5 | 41.3 |
| | 6 | 5.6 | 53.6 | 44.9 | -44.0 | 22.7 | 56.2 | 12.4 | -39.9 | **11.2** |
| TanzShed1 | 1 | 139.3 | -59.7 | 5.7 | **2.7** | -40.6 | 198.1 | -51.5 | -52.9 | -61.0 |
| | 2 | 2.6 | -89.6 | -17.6 | **-16.6** | -42.3 | -78.8 | -77.5 | -80.5 | -96.6 |
| | 3 | 705.5 | -23.5 | 85.6 | 40.3 | **-12.0** | 579.8 | -19.9 | -48.5 | -27.7 |
| | 4 | 13.3 | -80.2 | 44.7 | **14.0** | -14.4 | -51.7 | -62.8 | -78.7 | -93.7 |
| | 5 | 4.2 | 193.6 | **-11.7** | -20.3 | 68.4 | 56.5 | 34.2 | 33.9 | 148.4 |
| | 6 | 5.1 | 89.9 | 75.6 | 36.7 | 48.3 | 128.1 | 65.3 | **9.4** | 85.8 |
| TanzShed2 | 1 | 74.4 | -78.8 | **-16.2** | -31.0 | -51.4 | -54.9 | -74.7 | -70.6 | -78.9 |
| | 2 | 2.2 | -94.3 | -51.7 | **-45.2** | -52.6 | -74.6 | -79.3 | -70.9 | -94.0 |
| | 3 | 626.3 | -23.1 | **-2.2** | -8.1 | -36.3 | 39.6 | -44.3 | -49.8 | -41.8 |
| | 4 | 17.8 | -80.4 | -46.2 | -30.3 | **-3.1** | -24.9 | -56.4 | -52.7 | -84.2 |
| | 5 | 8.7 | 449.7 | -29.7 | **-10.3** | 118.3 | 294.0 | 230.9 | 155.0 | 555.4 |
| | 6 | 8.0 | 246.8 | **11.4** | 27.1 | 25.2 | 195.4 | 110.0 | 62.8 | 163.7 |

- (4) Please also consider reducing the number of sub-figures in Figure 4 by simply showing the correlation value for each of the sub-regions for a given dataset in one sub-figure hence you'll have a 5 columns single row figure. Not sure if it's necessary to show all the scatter diagrams. You could also just show correlation values for one reference dataset (such as CHIRPS) and show the difference in the correlation of other datasets and the correlation of the reference dataset.

Authors´ response:

- Thanks for the recommendation. Following your recommendation, we presented only the correlation (CC) for all the datasets in all the validation areas (1-21) explained in section 4.1 (page 14). As we explained above, we use this result to select the reference datasets (CHIRPS and ORH for rainfall and temperature). We use numbers to represent the validation areas for improving the figure resolution instead of the text (e.g., EthioShed1 as 1). We moved the scatterplots to the supplementary material (SF. 1) for providing further information such as biases.

[revised manuscript text omitted]

---

## Author Response (AR4)

Reply to comments by the editor on "Evaluation of Multiple Climate Data Sources for Managing Environmental Resources in East Africa"

Dear Authors,

Thank you for revising the manuscript. I would like to suggest one more minor but important (for clarity) comment. Could you use different terms for differentiating COORDEX-RCM and other RCMs? Right not use of the terms RCM and RCMs is confusing. May be use the terms "COORDEX-RCMs" and "Other RCMs"?

Dear Dr. Shukla,

Thank you very much for the acknowledgment and your recommendation for clarity. As we explained in page 9 lines 19-27 and page 10 lines 1-5, the RCMs used in this study are from the CORDEX ("*Historical data (control model runs) of the CORDEX RCMs are also used as a potential source for rainfall, T-max, and T-min data.*"). To make the term *RCM and RCMs* clear; "RCM" was used to indicate the individual RCMs and "RCMs" is the mean of all the RCMs used in this study. For clarity, the RCM is changed to "I-RCM". We hope this will make it clear.

[revised manuscript text omitted]